# Thermal fluctuations of the lipid membrane determine particle uptake into Giant Unilamellar Vesicles

Yareni A. Ayala[1], Ramin Omidvar [2,3], Winfried Römer [2,3,4] & Alexander Rohrbach [1,3] ✉

Phagocytic particle uptake is crucial for the fate of both living cells and pathogens. Invading particles have to overcome fluctuating lipid membranes as the first physical barrier. However, the energy and the role of the fluctuation-based particle-membrane interactions during particle uptake are not understood. We tackle this problem by indenting the membrane of differently composed Giant Unilamellar Vesicles (GUVs) with optically trapped particles until particle uptake. By continuous 1 MHz tracking and auto-correlating the particle's positions within 30μs delays for different indentations, the fluctuations' amplitude, the damping, the mean forces, and the energy profiles were obtained. Remarkably, the uptake energy into a GUV becomes predictable since it increases for smaller fluctuation amplitudes and longer relaxation time. Our observations could be explained by a mathematical model based on continuous suppression of fluctuation modes. Hence, the reduced particle uptake energy for protein-ligand interactions LecA-Gb3 or Biotin-Streptavidin results also from pronounced, low-friction membrane fluctuations.

For more than a billion years, thermal energy fluctuations have driven interactions of small structures and molecules, enabling them to find the best interaction state in time and space. For instance, particles undergo Brownian motion on scales down to sub-nanometer and sub-nanoseconds, or, in a similar way, biological membranes fluctuate at different wavelengths and frequencies. When a particle gets into contact with a (cellular) membrane, e.g., for binding, wrapping, and uptake, different membrane fluctuation modes interact with the particle on different timescales.

Membrane-particle interactions are important regulators for biological processes such as phagocytosis or pathogen invasion of cells, where particles are wrapped by the cell membrane and internalized into cells[1–4]. Although internalization is typically driven by membrane-cytoskeleton remodeling and protein activation, bacterial uptake is also possible by pure receptor-mediated adhesion forces and membrane wrapping without the interference of the cytoskeleton[5].

Given the complexity of biological cell membranes, it is often useful to simplify chemo-mechanical investigations by using cell-mimicking systems as giant unilamellar vesicles (GUVs)[6–8]. Such systems are composed of selectable phospholipid molecules that can be tailored to mimic basic structural characteristics, also found in biological membranes. These lipids define the elasticity and viscosity of the membrane, how the membrane fluctuates by thermal forces and if or how it engulfs a particle in contact with it.

Lipid membranes show undulations with different wavelengths, amplitudes, and frequencies, excited by the thermal forces from their viscous fluid environment[9]. These fluctuations seem to be stochastic at first glance, but follow physical boundary conditions, which can be

[1]Laboratory for Bio- and Nano-Photonics, Department of Microsystems Engineering - IMTEK, University of Freiburg, Georges-Köhler-Allee 102, 79110 Freiburg, Germany. [2]Faculty of Biology, University of Freiburg, Schänzlestraße 1, 79104 Freiburg, Germany. [3]Signalling Research Centres BIOSS and CIBSS, University of Freiburg, Schänzlestraße 18, 79104 Freiburg, Germany. [4]Freiburg Institute for Advanced Studies (FRIAS), University of Freiburg, Albertstraße 19, 79104 Freiburg, Germany. ✉e-mail: rohrbach@imtek.de

revealed by autocorrelation or spectral analysis. Any change in the surroundings of the membrane owing to an external mechanical perturbation, e.g., an indenting object, impacts its shape and hence on the spatiotemporal spectrum of its membrane fluctuations.

Typically, membrane reshaping is well described in terms of an elastic deformation characterized by the surface tension ($\sigma$) and the bending rigidity ($K$) of the membrane. However, the membrane's capability to be stretched ($\sigma$) and bended ($K$) must also define the uptake energy $G_{up}$ that an indenting object must overcome to penetrate lipid membranes. A particle's fluctuation behavior can be well described and measured by its fluctuation width or amplitude (leading to the stiffness $\kappa$) and its temporal fluctuation decay (damping) through energy dissipation (friction coefficient $\gamma$). The spectrum of the force kicks from the particle is transferred in a characteristic way to the vesicle membrane, depending on the membrane parameters $\sigma$ and $K$. This brings up the hypothesis that a thermal noise-driven particle in contact with giant vesicle probes the membrane properties, and thus can provide indications about the required uptake energies and forces.

Such measurements are possible with Photonic Force Microscopy (PFM), a combination of optical tweezers and fast thermal noise tracking, providing important fluctuation parameters for local interactions in addition to mean particle positions and forces[10–13]. Optical tweezers (OT) are a well-established micromanipulation technique, in which a highly focused beam traps small dielectric objects, typically microspheres (beads), used as force transducers in the range of Femto- to Piconewtons. The fastest and most precise particle tracking system is back-focal-plane interferometry, allowing three-dimensional position detection of the particle with nm precision at even MHz rates[14].

In this work, we use PFM to slowly push a 1 µm polystyrene (PS) sphere into DOPC-GUVs under different conditions to study the energies, forces and fluctuations involved during the uptake process. We record the diffusive motion of the particle at high spatiotemporal resolution (µs–nm) as it exerts a pushing force to deform and penetrate the membrane. From the position fluctuation data of the particle at short timescales, we measure and calculate the changes in the elastic ($\kappa$) and dissipative ($\gamma$) components of its surroundings resulting from the interaction with the membrane. We characterize the uptake process by measuring the energy cost of particle internalization as a function of the changes in the fluctuation parameters $\kappa$ and $\gamma$. We investigate how the GUV's material properties (defined by the surface tension $\sigma$ and the bending rigidity $K$), determine the uptake energetic cost. Finally, we explore the influence of the membrane-particle adhesion on the wrapping and internalization of the particle by using biotin-streptavidin linkages and bacterial lectin-glycan interactions.

## Results

### Force measurements and particle position tracking

A Photonic Force Microscope (PFM) is composed of an optical tweezers setup coupled to a fast 3D tracking system. A simplified sketch of the experimental setup is shown in Fig. 1a. The optical tweezers are used to trap and manipulate the particle in three dimensions. The trap is formed by an infrared laser beam (2 W Nd:GdVO4 solid-state laser, $\lambda = 1064$ nm TEM$_{00}$, Smart Laser Systems, Berlin, Germany) tightly focused by a NA = 1.2 water immersion objective lens (UPlanApo/IR 60XW, Olympus, Japan, lower objective lens in Fig. 1a) that spatially confines a PS bead in a liquid surrounding medium. The position fluctuations of the particle depend on the trap stiffness $\kappa_{i,opt}$, and the viscosity of the medium $\eta$ ($i = x,y,z$). The sample chamber containing the beads and the GUVs is mounted on a piezoelectric stage (PZ) for nanometer positioning control. The light coming directly from the focused beam interacts with the light scattered by the trapped particle, which is collected by a detection objective lens (W Plan-Apochromat ×63/ NA$_{det}$ = 1.0, Carl Zeiss, upper objective lens in Fig. 1a) forming an

interference pattern at its back-focal plane. Here, two quadrant photodiodes QPD$_{xy}$ and QPD$_z$, (InGaAs PIN photodiodes, G6849 series, Hamamatsu, Japan) record the particle position signals $S_i(t) = g_i \cdot b_i(t)$, which are approximately proportional (detector sensitivity $g_i \approx$ const) to the particle displacements $b_i(t)$, as shown in Fig. 1c. We achieve a maximum data acquisition rate of 2 MHz with amplifier electronics (miniSupply QUAD pre-amplifiers, TEM Messtechnik, Hannover, Germany) and two analog-digital data conversion boards (NiDAQ PCI-6110 and PCIe-6259, National Instruments, Austin, TX). For details see ref. 11. Sample imaging with bright field and fluorescence microscopy are combined with PFM particle trapping and tracking. We imaged and investigated both flaccid and tense GUVs (see Supplementary Note 1 and Supplementary Movies 1, 2). For uptake experiments, we approached the GUV towards the stationary optical trap using the PZ stage as illustrated in Fig. 1a and Supplementary Figs. S1, S2. Within typical window sizes $\Delta t = 0.1$ s (used for subsequent equilibrium fluctuation analysis), a 10 nm stage shift brings the particle negligibly little out of thermal equilibrium. The GUV was fixed at the bottom of the sample between two large glass beads. Depending on the deformability of the GUV, we varied the optical trap stiffness with laser powers ranging from 3 mW to 60 mW within the sample equivalent to lateral stiffnesses $\kappa_{opt,xy} = 4$ pN/µm to 75 pN/µm, respectively. We used a low power range to avoid phase transitions or damage on the GUVs during the experiments[15], which lasted less than 100 s. The interaction between particle and membrane results in mean bead displacements in all three directions with respect to the trap center ($b_{i0} = 0$), but also changes the bead fluctuations. However, since uptake experiments were performed along the y-direction, the interaction with the GUV mainly affects the signal $b_y$ whereas has a lesser effect on $b_x$ and $b_z$. Therefore, we focus our analysis in this study on the position fluctuations mainly along the y-direction (see Supplementary Movies 3–6).

### Thermal particle fluctuations for different indentation depths

An optically trapped particle (radius $R_b = 0.5$ µm) is subject to restoring, approximately linear optical forces $F_{opt,i} \approx \kappa_{opt,i} \cdot b_i$ and dissipative fluid friction forces $F_{\gamma,i} \approx \gamma_{bd} \cdot \dot{b}_i$ described by the trapping stiffness $\kappa_{opt,i}$ and the fluid drag or friction coefficient $\gamma_{bd} = 6\pi\eta_{fl}R_b$ ($\eta_{fl}$ is the fluid viscosity). $b_i(t)$ is the time-varying displacement from the trap center, $\dot{b}_i$ the corresponding velocity. In contact with the vesicle membrane at distance $d = 0$, both the stiffness and the friction coefficient change with the particle's indentation depth $d$ into the vesicle, such that in one dimension the stiffnesses $\kappa_{tot}(d) = \kappa_{opt} + \kappa_{mem}(d)$ and friction constants $\gamma_{tot}(d) = \gamma_{bd} + \gamma_{mem}(d)$ add up according to a parallel connection in equilibrium, an approximation valid for a system of infinite size. This arrangement is illustrated in Fig. 1b, where one trap position is outside the GUV (negative distance $d_0$), with $\kappa_{tot}(d_0 \ll 0) = \kappa_{opt}$ and $\gamma_{tot}(d \ll 0) = \gamma_{bd}$ and where two trap positions are inside the GUV at $d_1$ and $d_2$. The membrane fluctuations—as sketched in the same figure—are defined by the surface tension $\sigma$ and bending rigidity $K$. These two membrane physiological parameters manifest different wave-like deformation modes $n$ with different spatial frequencies $q_n$ and temporal frequencies $\omega_n$. Each mode has an intrinsic friction $\gamma_{mem}$ and elasticity $\kappa_{mem}$ and all modes add up independently (in series) as indicated by the inset of Fig. 1a.

The three exemplary trap positions $d_0$, $d_1$ and $d_2$ (indicated by gray arrows in Fig. 1c) reveal different mean displacements $\bar{b}(t)$ and amplitude fluctuations $\delta b(t)$, displayed for all 3 directions in red, green and blue over the 90 s time course of a complete experiment. Time windows of 150 ms length with their corresponding decaying autocorrelation (AC) functions are shown in Fig. 1d for the same distances $d_0$, $d_1$ and $d_2$. Three insets sketch the membrane free energy $G_{mem}$ (in green) relative to the harmonic optical trapping potential $V_{opt}$ (in red) and the resulting particle displacement.

The question is how the membrane fluctuations can be extracted from the bead's autocorrelation functions $AC(\tau)$ providing $\kappa_{opt}$ and $\gamma_{tot}$

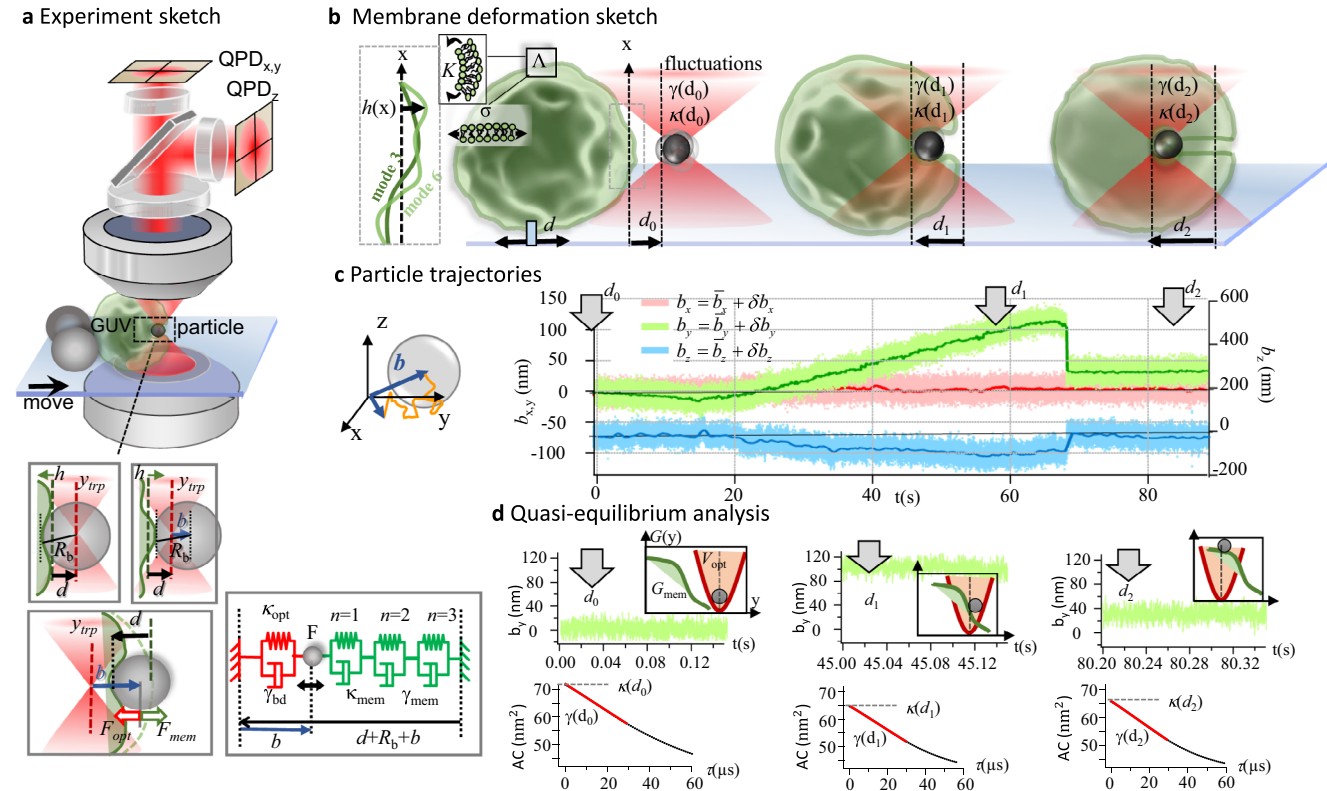

**Fig. 1 | Particle trapping and tracking at different internalization stages. a** Setup sketch: A polysterene particle (radius $R_b = 0.5\,\mu m$) is optically trapped by an IR-laser focus. A giant unilamellar vesicle (GUV) is immobilized by $20\,\mu m$ silica beads at the bottom of the coverslip, which is moved toward the trapped particle (y-direction). The particle positions are tracked at 1 MHz by back-focal plane interferometry using two quadrant photodiodes $QPD_{x,y}$ and $QPD_z$ to be converted into particle positions $b_x, b_y$, and $b_z$ relative to the fixed trap center $y_{trp}$. Inset: the bead-membrane surface distance $d$ varies with the particle's center position $b_y = b$ relative to $y_{trp}$ as a result of optical force $F_{opt}$ and membrane force $F_{mem}$; $h$ represents the membrane shape deformation. The system is modeled by Kelvin Voigt elements (KVE): the optical trapped bead (with fluctuation parameters trap stiffness $\kappa_{opt}$ and bead friction $\gamma_{bd}$) is connected in parallel to the total membrane KVE (with membrane stiffness $\kappa_{mem}$ and membrane friction $\gamma_{mem}$), which itself consists of serial KVE connections representing independent membrane modes ($n = 1, 2, 3,...$). **b** Sketch of three different stages of induced uptake into a GUV with membrane

fluctuations defined by the surface tension $\sigma$ and the bending rigidity $K$. At $t = 0$ a flaccid GUV is approached to the trapped particle at the surface distance $d_0$. After contact, the membrane partially wraps the particle at $d_1$. At $d_2$ the particle inside the GUV is completely wrapped forming an inward membrane tether. **c** Three trajectories $b_x(t), b_y(t)$ and $b_z(t)$ represent the relative displacement of the particle center from the trap center $y_{trp}$. The particle fluctuates by $\delta b_{x,y,z}(t)$ (bright lines) around a mean value $\bar{b}_{x,y,z}(t)$ (dark solid curves) corresponding to the smooth of raw data over $\Delta t = 100\,ms$. The gray arrows link the exemplary displacements to the internalization stages in b. **d** Millisecond segments of $b_y$ (top graphs) corresponding to the distances $d_0$, $d_1$ and $d_2$ used in the autocorrelation (AC) analysis over subsequent 150 ms time windows. Insets: Interaction of harmonic optical trapping potential $V_{opt}$ and membrane free energy $G_{mem}$, $AC(\tau)$ functions are linearly fitted at $\mu s$ scales (red solid line) to obtain the changing fluctuation parameters, i.e. the stiffness $\kappa(d)$ and friction $\gamma(d)$.

for different indentation distances $d$ and for different membrane properties $\sigma$ and $K$. By unraveling this relationship, we might uncover the role the thermal membrane fluctuations play for the particle uptake energy and probability.

## Thermal fluctuations of different membrane modes

The goal is to explain and interpret the experimental results, by developing a simplified theoretical description based on linear systems theory in thermal equilibrium, which is approximately given by the stepwise approach of the bead (10 nm steps and 0.1 s dwell time).

Since the fluctuation amplitudes and relaxations of a quasi-planar lipid membrane (Monge parametrization) are nearly the same than that of a spherical giant vesicle[9] (see Supplementary Fig. S3), we use the quasi-planar symmetry (Monge parametrization) for the ease of writing. The GUV has a radius $R_g$, its membrane deforms by a height function $h(\mathbf{r})$ according to a bending modulus (or rigidity) $K$ and a surface tension $\sigma$. Bending and stretching the membrane costs energy (and force), which are proportional to $K$ for bending and to $\sigma$ for stretching lipids relative to each other, as described by the Helfrich free energy $G_{mem} = \frac{1}{2}\int_A \left[ K \cdot (\nabla^2 h(\mathbf{r}))^2 + \sigma \cdot (\nabla h(\mathbf{r}))^2 \right] d^2 r$[16,17]. The membrane

deformations $h(\mathbf{r})$ within an area $A$ can be composed by a sum of $N$ Fourier modes, $h(\mathbf{r}) = \frac{1}{A} \sum_{n \leq N} h(\mathbf{q}_n) \cdot \exp(i\mathbf{q}_n \mathbf{r})$, such that the membrane deformation energy is $G_{q,mem} = \frac{1}{2} \sum_{q=1}^{N} \left[ K \cdot \mathbf{q}^4 + \sigma \cdot \mathbf{q}^2 \right] \cdot |h(\mathbf{q})|^2$[9,18,19]. Here, $q_0 = \pi/R_g$ is the smallest and $q_{max} = \pi/R_b$ the largest wave number $q = |\mathbf{q}|$ considered; all modes scale as $q_n = n \cdot q_0$ (in a typical experiment $n \leq N_{mx} \approx R_g/R_b \approx \frac{13\,\mu m}{0.5\,\mu m} = 26$). Each fluctuation mode with mode number $n$ can be described by a damped spring (Kelvin Voigt element, KVE), with elasticity $\kappa_{mq}(n) = q_0^2(K(nq_0)^4 + \sigma(nq_0)^2) = K \cdot n^4 q_0^2 + \sigma \cdot n^2$ (pN/μm) and damping (friction) coefficient per area $\gamma_{mq}(n) \approx n \cdot 4\eta q_0 = n \cdot 4\pi\eta/R_g$ (s pN/μm³)[17].

The motions $b_y(t) = b(t)$ in the radial $y$-direction of a bead in contact with the membrane are strongly controlled by the (local) spatial GUV deformations $h(\mathbf{r}) = b$ and by the temporal fluctuations $h(t) = b(t)$ as illustrated in Fig. 1a, b. While the membrane deforms around $h_0 = 0$ by $h(t)$, the bead center is displaced around $y_{trp}$ by $b(t)$. We approximate both the optical force $F_{opt} = \kappa_{opt} \cdot b$ and the elastic membrane force $F_{mem}(b,d) \approx \kappa_{mem}(d) \cdot b$ to be linear for all mean distances $d$ between bead surface and membrane.

Based on the total free energy $G(\mathbf{b}) = V_{opt}(\mathbf{b},d) + G_{mem}(\mathbf{b})$ (with optical potential $V_{opt}(b) = \frac{1}{2}\kappa_{opt} \cdot (b - d)^2$ along the $y$-direction), the corresponding overdamped equation of motion of the bead position

$b(t)$ relative to center of the trapping beam can be approximated as

$$\int_{-t_{min}}^{t} (\gamma_{bd} + \gamma_{mem}(t',d)) \cdot \dot{b}(t-t')dt' + (\kappa_{opt} + \kappa_{mem}(d)) \cdot b(t) \approx F_{th}(t) \quad (1)$$

We assume that the friction force $\gamma_{bd} \cdot \dot{b}$ the particle experiences increases by the distance-dependent force $\gamma_{mem}(d) \cdot \dot{b}$ due to membrane contact and wrapping. The frictional terms consider time memory from $-t_{min} \rightarrow -\infty$, which effectively corresponds to the period within the particle reaches equilibrium. We further assume uncorrelated white thermal noise for both the membrane fluctuations $\sum_q F_{th,q}^{mem}(t,d)$ and the bead fluctuations $F_{th}^{bd}(t,d)$, which are put together in $F_{th}(t) \approx F_{th}^{bd}(t,d) + F_{th}^{mem}(t,d)$ independent of $d$, such that $AC[F_{th}(t)] = 2\gamma_{tot}k_BT \cdot \delta(t-t')$.

$\kappa_{mem}(d)$ changes with $d$, because of changing membrane deformations but also because the displacement of one potential is related to the other (insets of Fig. 1d).

By Fourier transforming Eq. (1) in time, we find $\tilde{b}(\omega)((i\omega\gamma_{bd} + \kappa_{opt}) + (i\omega\gamma_{mem} + \kappa_{mem})) = \tilde{F}_{th}(\omega)$ or correspondingly the frequency dependent displacement of the bead can be expressed by the response functions $\alpha_{bd}$ and $\alpha_{mem}$ of bead and membrane:

$$\tilde{F}_{th}(\omega) = \tilde{b}(\omega)\left(\frac{1}{\alpha_{bd}(\omega)} + \frac{1}{\alpha_{mem}(\omega)}\right) = \tilde{b}(\omega)\left(\frac{1}{\alpha_{bd}(\omega)} + \frac{1}{\sum_N \alpha_{qm}(\omega)}\right)$$
$$= \tilde{b}(\omega)\frac{1}{\alpha_{tot}(\omega,N)} \quad (2)$$

As sketched in Fig. 1a bottom, the damped springs of the membrane and the optical trap form a parallel connection, such that response functions $\alpha_{bd}$ and $\alpha_{mem}$ add up inversely. As expressed in Eq. (2), all membrane deformation modes form a serial connection such that their response functions $\alpha_{qm}$ add up directly, where $\alpha_{qm}(\omega)^{-1} = \kappa_{qm} + i\omega\gamma_{qm} = (Kn^4q_0^2 + \sigma n^2) + i\omega(4\eta n q_0)$. The latter rule corresponds to an addition of mode displacements $\sum_n \tilde{b}_n(\omega) = \tilde{F}_{th}(\omega)\sum_n \alpha_{qm}(\omega)$.

From Eq. (2) one obtains the power spectral density $|\tilde{b}(\omega)|^2 = |\alpha_{tot}(\omega,N)|^2 |\tilde{F}_{th}(\omega)|^2$

$$|\tilde{b}(\omega)|^2 = \left| \frac{1}{(\kappa_{opt} + i\omega\gamma_{bd}) + \left(\sum (\kappa_{qm} + i\omega\gamma_{qm})^{-1}\right)^{-1}} \right|^2 \cdot 2\gamma_{tot}k_BT \quad (3)$$

Further information about response functions can be found in Supplementary Note 4.

The equipartition theorem $\langle|b|^2\rangle = k_BT \cdot \alpha(\omega = 0)$ used for thermal equilibrium ($\omega \rightarrow 0$) provides the mean square fluctuations of the bead $\langle|b(n)|^2\rangle$ connected to membrane mode $n$ or $\langle|b(N)|^2\rangle$ connected to the sum of all modes, both determined by optical and membrane forces

$$\langle|b(n)|^2\rangle = \frac{k_BT}{\kappa_{opt} + (Kn^4q_0^2 + \sigma n^2)}; \langle|b(N)|^2\rangle = \frac{k_BT}{\kappa_{opt} + \left(\sum_{n=1}^{N}(Kn^4q_0^2 + \sigma n^2)^{-1}\right)^{-1}} \quad (4)$$

This dependency is shown in Fig. 2a, where the mean fluctuation widths $\langle|b(n)|^2\rangle^{1/2}$ of individual modes $n$ are shown in dashed lines and the widths $\langle|b(N)|^2\rangle^{1/2}$ of the mode sum $N$ are shown in solid lines. Without bead and optical trap (red curve), the total fluctuation width increases with increasing mode sum $N$ (solid red line), while the fluctuation width of individual modes decreases with increasing mode number $n$ (dashed red line). For the relevant case of a bead in contact with the vesicle membrane (blue curve), the optical trap restricts the fluctuation width for an increasing mode sum $N$ (solid blue line),

whereas individual modes fluctuate less with increasing mode number $n$ (dashed blue line).

The autocorrelation function $AC[b(t)] = AC(\tau)$ is the inverse Fourier transform of Eq. (2) and can be computed from the trajectories $b(t)$ as displayed in Fig. 1d. The autocorrelation $AC[b(t,N)] \approx \langle|b(N)|^2\rangle \cdot \exp(-\tau/\tau_c(N))$ is exponentially decaying for linear restoring forces. On short-time delays $\tau < \tau_s = 30\mu s \ll \tau_c$, we use a linear approximation $AC(\tau) \approx k_BT/\kappa_{tot} \cdot (1 - \tau/\tau_{tot})$, which is justified even for higher membrane modes as demonstrated in Fig. 2d (see narrow shaded area for $\tau < 50 \mu s$):

$$AC(\tau, N) = \frac{k_BT}{\kappa_{tot}(N)} \cdot \exp\left(-\tau \cdot \frac{\kappa_{tot}(N)}{\gamma_{tot}(N)}\right)$$
$$\approx \frac{k_BT}{\kappa_{tot}(N)} - \frac{k_BT}{\gamma_{tot}(N)} \cdot \tau \quad (5)$$

Since every mode decays independently of the others, the autocorrelation functions can be added up, which is significantly easier and better to analyze experimentally on very short timescales ($\tau \ll \tau_c$).

Both the mean square fluctuation width $AC(\tau = 0)$ and the relaxation time $\tau_c(N)$ change with the indentation depth $d$ of the particle. By measuring the equilibrium value $AC(0, d)$, we obtain $\kappa_{tot}(d)$ and by measuring the slope $-k_BT/\gamma_{tot}(d)$ on short timescales we obtain $\gamma_{tot}(d)$ for each depth (trap position) $d$.

By analyzing the autocorrelation decay in Eq. (5) and by using the mean fluctuation widths of Eq.(4), we identified the total autocorrelation time can be expressed by the vesicle radius $R_g$, the viscosity $\eta$, the bending modulus $K$, the membrane tension $\sigma$, and the mode sum $N$ according to

$$\tau_c(N, n_0) \approx \left(\gamma_{bd} + \left(\frac{R_g}{4\pi\eta}\sum_{n_0}^{N}\frac{1}{n}\right)^{-1}\right) \bigg/ \left(\kappa_{opt} + \left(\sum_{n_0}^{N}\frac{1}{Kn^4q_0^2 + \sigma n^2}\right)^{-1}\right) \quad (6)$$

Figure 2b displays the decay of the relative viscous drag $\gamma(N)/\gamma_{bd}$ and of the relative stiffness $\kappa(N)/\kappa_{opt}$ (i.e., relative to $\gamma_{bd}$ and $\kappa_{opt}$ without membrane) experienced by the bead at the membrane. Whereas the stiffness decays hardly with mode sum $N$, the total friction decreases stronger with $N$. This is the case for the membrane only condition (red curve), but also for the bead attached to the membrane of two different vesicles (light blue vesicle with radius $R_g = 7.5 \mu m$ and blue with $R_g = 13 \mu m$).

The mode relaxation times $\tau_c(N)$ in Eq. (6) are illustrated in Fig. 2c. The lowest mode relaxes on $\tau_c(N=1)$, i.e. on the order of a few milliseconds, but $\tau_c(N>1)$ becomes shorter with increasing number $N$ of (higher) modes (dashed curves). Relaxation times $\tau_c(N)$ are shorter for smaller vesicles (light blue) than for larger GUVs (blue). As defined in Eq. (6) and more important for this study, it is also possible to decrease the number of modes by suppressing lower modes, i.e. by increasing the lower mode number $n_0$. The solid lines show an increase in the relaxation time, when lower modes are suppressed successively. Supplementary Fig. S4 is further illustrating this effect for typical parameters.

One can consider the dependency of the fluctuation parameters $\kappa_{mem}(d)$ and $\gamma_{mem}(d)$ on the distance $d$ introduced in Eq. (1), which changes with the maximum number $N_{mx}(d)$ of modes, or, relevant to the study with the lowest mode $n_0(d) = d \cdot (N_{mx} - 1)/d_{up}$ such that the relaxation time is $\tau_c = \tau_c(N, n_0)$:

$$\tau_c(d < d_{up}) = \frac{\gamma(d)}{\kappa(d)} \approx \tau_c\left(N_{mx}, \left(d \cdot (N_{mx} - 1)/d_{up}\right)\right) \quad (7)$$

For the approximation on the right we have assumed that the number of fluctuation modes decreases linearly with the indentation depth $d$ of the bead for a maximum number of modes $N_{mx} = R_g/R_b$.

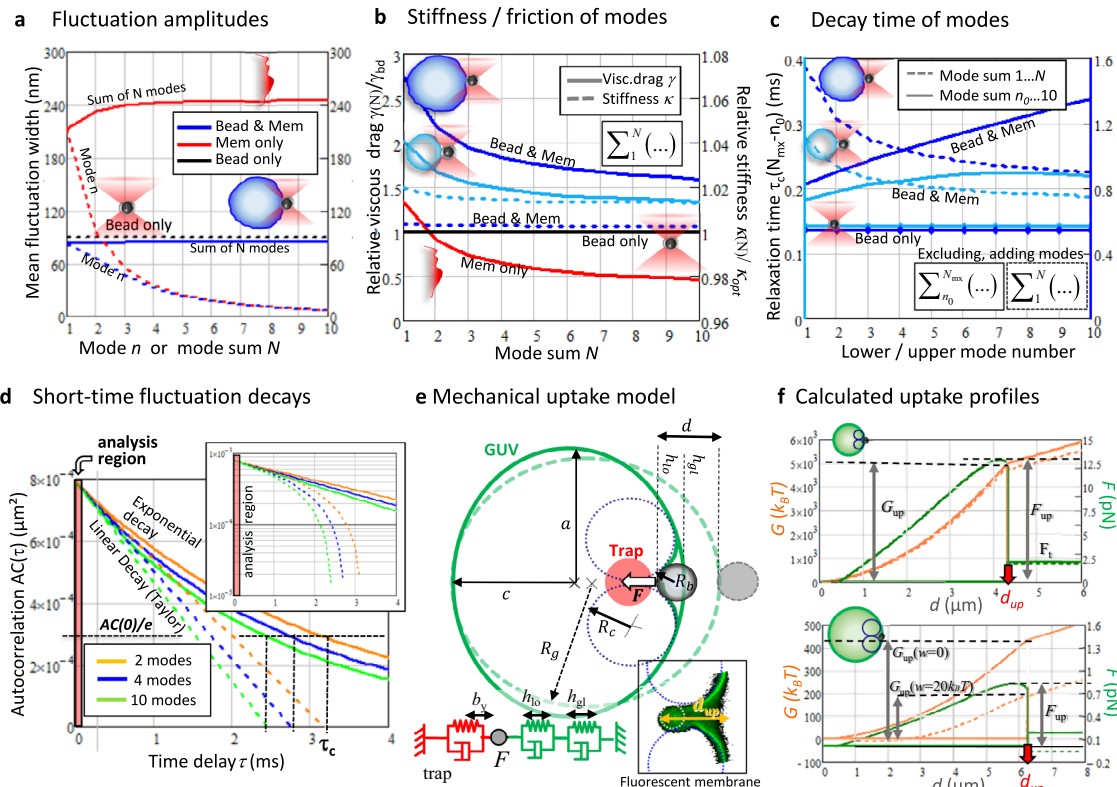

**Fig. 2 | Coupling between particle fluctuations and membrane fluctuations.**
**a** Change of the mean fluctuation width $|b_n| = \langle|b_n|^2\rangle^{1/2}$ of single modes $n$ with
wavelength $2R_g/n$ (dashed curves) and $|b_N| = \langle|b(N)|^2\rangle^{1/2}$ of the sum of $N$ inde-
pendent modes (solid curves). Total fluctuation widths corresponding to mem-
brane only (Mem only in red), membrane with trapped bead (Bead & Mem in blue)
and bead only (black) with $R_g = 10\,\mu m$, $K = 20\,k_B T$, $\sigma = 0.08\,pN/\mu m$, and
$\kappa_{opt} = 5\,pN/\mu m$. **b** The membrane viscous drag $\gamma$ ($\gamma$ with/without trapped particle)
decays for a larger number of modes $N$ whereas the stiffness $\kappa$ remains approxi-
mately constant with $N$. $\gamma$ and $\kappa$ are relative to those of the bead only (normalized by
$\gamma_{bd}$ and $\kappa_{opt}$). Colors: Tense vesicle in light blue ($R_{g1} = 7.5\,\mu m$, $\sigma_1 = 300\,k_B T/\mu m^2$ and
$K_1 = 12\,k_B T$), flaccid vesicle in blue ($R_{g2} = 13\,\mu m$, $\sigma_2 = 15\,k_B T/\mu m^2$ and $K_2 = 1.6\,k_B T$)
and membrane only in red. **c** The autocorrelation times $\tau_c(N) = \gamma(N)/\kappa(N)$
describing the bead relaxation in contact with the membrane decrease with the
number $N$ of higher modes added (dashed curves), $AC$ times $\tau_c(N - n_0)$ increases
for subsequent exclusion of lower modes, i.e. increasing of lower mode number $n_0$

(solid curves), parameters are set as in b. **d** The total autocorrelation time $AC(\tau, N)$
decays nearly exponentially (solid lines), shown by a linear and a semi-logarithmic
(inset) plot for the sum of $N = 2$ (yellow), 4 (blue) and 10 (green) modes. The linear
Taylor approximation (dashed lines) of the $AC$ coincides for short-time delays of
$\tau < 50\,\mu s$ (red shaded region). The $\tau_c$ corresponds to a decay of $AC(0)$ by $1/e$.
**e** Deformation of a spherical vesicle (GUV) with radius $R_g$ to an oblate ellipsoidal
vesicle with half-axes $a$ and $c$. The global ($h_{gl}$) and local ($h_{lo}$) deformations result
from the displacement $b_y$ of an optically trapped particle with radius $R_b$. Locally,
the deformation is described by a torus with minor radius $R_c$. Inset: Membrane tube
formation at the uptake distance $d_{up}$. **f** Calculated force $F(d)$ (green) and energy
$G(d)$ (orange) profiles for the tense and flaccid vesicle, parameters are set as in b.
The uptake force $F_{up} = F(d_{up})$ and uptake energy $G_{up} = G(d_{up})$ are taken at $d_{up}$,
where $F(d)$ drops to tether force $F_t$. The dashed energy profiles indicate the
influence of the adhesion force from surface energy density $w = 20\,k_B T$ between
bead and membrane surfaces.

The relaxation time and the number of modes ($N_{mx} - n_0$) change with
$d$ until the particle is wrapped into the membrane for $d < d_{up}$. The
uptake distance $d_{up}$ is defined in the following section.

## Membrane deformation by a particle

Using fluorescence microscopy, one can see two different types of
GUV deformations[10]. As sketched in Fig. 2e, there is a global defor-
mation from the round GUV with radius $R_g$ to an ellipsoidally deformed
GUV (with half-axes $a$ and $c$) caused by force $F_{opt}$ of the trapped par-
ticle. This deformation, defined by the mean free energy $G^{glo}(h_{gl})$, can
be quantified by the global ($gl$) indentation distance $h_{gl}$. In addition,
there is a local ($lo$) deformation caused by the bead, which generates a
toroidal membrane indentation with torus minor radius $R_c$. Two circles
with appropriate radii $R_c$ fit well to fluorescence indentation profiles of
the GUV (Fig. 2e bottom).

The local indentation distance is denoted as $h_{lo}$ and is defined by
the local free energy $G^{loc}(h_{lo})$ of the membrane. The corresponding
damped (nonlinear) springs form a serial connection with each other,
but a parallel connection with the optical trap, as outlined in Fig. 2e
bottom. Therefore, the total free energy $G_{mem}(d)$ is explored by the

addition of the local and global indentation lengths $h_{lo} + h_{gl} = d(h_{lo})$
according to

$$G_{mem}(h_{lo} + h_{gl}) = \begin{cases} G^{loc}(h_{lo}) + G^{glo}(h_{gl}) & \text{if } d < d_{up} \\ G_{tube}(d(h_{lo} + h_{gl})) & \text{if } d > d_{up} \end{cases} \quad (8)$$

The indentation function $d(h_{lo}) = h_{lo} + h_{gl} = h_{lo} + z(h_{lo})h_{lo}$ can
be found by the minimum of the potentials $(G^{loc}(h_{lo}) + G^{glo}(h_{gl})) \to$ min
or for the zeros $z(h_{lo})$ of the corresponding force difference
$F^{loc}(h_{lo}) - F^{glo}(h_{gl}) = 0$, such that $F^{loc}(h_{lo}) = F^{glo}(z(h_{lo}))$, as
required for a serial connection of forces. As introduced in ref. 10, the
Helfrich free energy in Eq. (8) can be split into global and local terms
for stretching and bending, such that
$G_{mem}(d(h_{lo})) = (-G_{ad}(h_{lo}) + G^{loc}_{ben}(h_{lo}) + G^{loc}_{str}(h_{lo})) + (G^{glo}_{ben}(h_{gl}) + G^{glo}_{str}(h_{gl}))$
(see Supplementary Note 5). Here, $G_{ad}(h_{lo}) = 2\pi \cdot w \cdot R_b \cdot h_{lo}$ is the
adhesion energy with surface energy density $w$, which will become
important for specific biological binding partners used in the study.

The profiles for the membrane energy and force during particle
indentation have been calculated based on Eq. (8) and are shown in

Fig. 2f for a smaller tense GUV (top) and a larger flaccid GUV (bottom). The values for the optical trap and the membrane parameters were taken from experimental results (as discussed later). Both energy and force profiles reveal nonlinearities and steps resulting from the composition of the (4+1) different energy components $G_{ben}^{loc} + G_{str}^{loc} + G_{ben}^{glo} + G_{str}^{glo} - G_{ad}$. The dashed orange lines describe the energy profiles lowered by an adhesion energy $G_{ad}(h_{lo})$ with $w = 20\,k_BT/\mu m^2$, which halves the uptake energy $G_{mem}(d_{up}) - G_{ad}(d_{up})$ of the specific uncoated particle into the flaccid vesicle. For a complete wrap of a spherical $1\,\mu m$ bead, we read $d_{up} = 4.7\,\mu m$ from Fig. 2f and find $G_{ad}(d_{up}) = \pi \cdot w \cdot 1\mu m \cdot 4.7\mu m = 443\,k_BT$. This way one can compare different particle-membrane interactions through $w = G_{ad}(d_{up})/(2\pi \cdot R_b \cdot d_{up})$ for GUVs of the same size and chemical composition.

## Energies and particle fluctuations at uptake

At the critical indentation distance $d_{up}$, the optical force overcomes the maximum membrane stall force, $F_{opt}(b_{max}) > F_{mem}(d_{up})$ and a membrane tube is formed inside the vesicle, while the GUV recovers its spherical shape. The critical force $F_{up} = F_{mem}(d_{up})$ is called the uptake force and the corresponding critical energy $G_{up} = G_{mem}(d_{up})$ is called the uptake energy (see Fig. 2f).

The required energy to take up a particle is related to the uptake probability and depends on the membrane properties described by bending rigidity $K$ and surface tension $\sigma$, resulting in a specific change of the membrane fluctuations. These changes are transferred to the particle fluctuations, which are described by $\kappa_{opt} + \kappa_{mem}(N)$ and $\gamma_{bd} + \gamma_{mem}(N)$ according to Eq. (6). Following Eqs. (4) and (7), the mean amplitude relaxation times of the fluctuations change with distance. Therefore, the following relation between uptake energy $G_{up} = G_{mem}(\tau(d_{up}))$ and auto-correlation time is of great relevance for our study:

$$G_{up}(N,K,\sigma) = G_{mem}\left(\gamma_{mem}\left(d_{up}\right)/\kappa_{mem}\left(d_{up}\right)\right) = G_{up}\left(\gamma_{up}, \kappa_{up}\right) \quad (9)$$

here $\tau_{up} = \gamma_{up}/\kappa_{up}$, $\kappa_{up} = \kappa_{up}(N(d_{up}),K,\sigma)$ and $\gamma_{up} = \gamma_{up}(N(d_{up}),K,\sigma)$ are the maximum stiffness and friction right before particle uptake.

For penetrations $h > d_{up}$, the free energy of the tube is $G_{tube}(h,R) = \frac{\pi}{R}K \cdot h + 2\pi\sigma Rh - h \cdot F_{opt}$. This energy is minimized $\left(\frac{\partial G}{\partial h} = \frac{\partial G}{\partial R} = 0\right)$ in $R$ and $h$ for the tether force $F_t = F_{opt}$ and the tether radius $R_t$[20]:

$$F_t(K,\sigma) = 2\pi \cdot \sqrt{2K\sigma} = 2\pi \cdot \frac{K}{R_t(K,\sigma)} \quad (10)$$

with radius $R_t = \sqrt{K/2\sigma}$ and characteristic length $\Lambda = \sqrt{K/\sigma} = \sqrt{2} \cdot R_t$, such that $F_t(K,\sigma) = \sqrt{8}\pi \cdot K/\Lambda(K,\sigma)$. The tube diameter $2R_t$ can be estimated from experiments with high-resolution microscope. By measuring the tether force $F_t$, the two membrane parameters $K = \frac{1}{2\pi}F_t \cdot R_t$ and $\sigma = \frac{1}{4\pi}F_t/R_t$ can be determined as well. The parameter $\Lambda$ describes the shape and the amplitude of the fluctuations: $\Lambda < 0.3\,\mu m$ describes tense vesicles revealing minimal bending undulations (large $K$) and fluctuation amplitudes. $\Lambda > 0.3\,\mu m$ describes flaccid vesicles, which can easily bend ($K$ small) and show pronounced fluctuation amplitudes (see Supplementary Movies 1–4).

## Measured profiles for force and energy during particle uptake

The experiments on particle uptake induced by optical forces were performed on DOPC-GUVs with different mechanical properties. Using fluorescence images of Texas Red-DHPE lipid membranes, Fig. 3a illustrates the results obtained for the indentation of an uncoated, $1\,\mu m$ bead into a tense vesicle (left side) and flaccid vesicle (right side). During the 10 nm stepwise approach of the trapped particle toward the GUV, the force fluctuates by $\delta F(d)$ (bright green lines) around a mean value $\langle F\rangle(d)$ shown by the smoothed, dark green curve. To establish thermal equilibrium, we waited for 0.1 s (dwell time) and measured

100.000 data points at each position. A slight decay in force indicates the first contact of the particle with the fluctuating membrane and can be explained by light scattering at the GUV (stage I in Fig. 3b). Once contact between membrane and particle has been established at $d = 0$ (visible in PFM tracking data, not in videos), the force along the indentation direction $y$ increases linearly with distance $d$, see stage II in Fig. 3b. In this stage, as the particle indents the GUVs, the membrane undulations are flattened (see Supplementary Movies 3, 4) and the particle is gradually wrapped producing an inward protrusion, as shown in the corresponding fluorescent images in Fig. 3a. The bead is outlined by a dashed white circle and the torus ring by a red dashed circle, which revolves around the white circle.

When a maximum area of the particle is wrapped by the membrane, the force becomes maximal at the uptake distance $d = d_{up}$ (second vertical dotted lines in Fig. 3b). The increase in the force from the contact point at $d = 0$ to the value at $d = d_{up}$ is the uptake force $F_{up} = F(d_{up}) - F(0)$. Immediately after $F_{opt} > F_{up}$, the GUVs exhibit a shape transition from a catenoidal-like shape to a tubular one (stage III). The force drops to a constant value $F_t$ required to produce an inner tube of radius $R_t$[20]. The transition from $F_{up}$ to $F_t$ is used as a reference to consider an uptake event as successful. Otherwise, the uptake experiment is frustrated, meaning that the optically driven particle could not overcome the membrane tension $\sigma$. Interestingly, we have measured tube transition times from $F_{up}$ to $F_t$ occurring in a temporal range from a few seconds to less than $1\,\mu s$, as shown on the right of Fig. 3f (shaded area). The radius $R_t$ of the formed tube and the force $F_t$ needed to elongate it, are the only two parameters required to obtain the material properties of the membrane, i.e. the bending rigidity $K$ and the membrane tension $\sigma$. $F_t$ is readily obtained from the force-extension curves in Fig. 3b. The tube radius $R_t$ is estimated from the fluorescent images of the tubes by fitting a Gaussian function to lateral linescans (see Fig. 3f). The deformation energy profile $G(d)$ was calculated as the integral of the force–distance curves $G(d) = \int_0^d F(y)\mathrm{d}y$ and is plotted at the right axis of Fig. 3b (orange colors). Correspondingly, the uptake energy is $G_{up} = G(d_{up}) - G(0)$. For the tense vesicle in the example of Fig. 3b, we measured $G_{up} = 5066\,k_BT$ at $d_{up} = 5.2\,\mu m$ contrasting with the nearly 10 times smaller $G_{up} = 588\,k_BT$ at $d_{up} = 7.3\,\mu m$ for the flaccid vesicle. After a successful uptake in stage IV, the energy increases linearly with the distance as a result of the constant force $F_t$ at which the tube is extended. Table 1 summarizes the main parameters characterizing the optically trapped particle fluctuation and membrane properties of DOPC vesicles.

## Measured profiles for fluctuation parameters during particle uptake

The changes in the particle's viscous drag $\gamma_{tot}(d)$, elasticity and relaxation time $\tau_c(d) = \gamma_{tot}/\kappa_{tot}$ of the particle as it is wrapped by the DOPC lipid membrane are shown in Fig. 3c, d for a tense and a flaccid vesicle over the indentation distance $d$. The values of $\gamma_{tot}(d < 0)$ and $\kappa_{tot}(d)$ in stage I are set by the optical trap stiffness, which is tuned depending on the flaccidity of the membrane to be penetrated. Shortly before reaching the membrane, there is a slight increase in the viscous drag component $\gamma_{tot}(d)$ probably due to the hydrodynamic coupling between the fluctuating lipid membrane and the approaching particle[11]. At the beginning of stage II, (Fig. 3c, left), the changes in friction and stiffness are minor for the tense vesicle. At about $d = 3\,\mu m$ both parameters increase steeply reaching their maximum value at $d_{up} = 5.2\,\mu m$, just before uptake occurs. From $d = 0$ to $d = d_{up}$ the viscous component $\gamma_{tot}(d)$ is increased 1.8-fold its initial value while the elastic component $\kappa_{tot}(d)$ 1.2-fold. The small increase in $\kappa_{tot}$ coincides with the hardly visible reduction in fluctuation width marked in Fig. 3b. The corresponding relaxation time $\tau_c(d)$ increases from 0.15 ms to 0.22 ms during uptake. On the other hand, the indentation of the flaccid membrane (Fig. 3c right) affects only the viscous component leading to a 1.5-fold increase, while the fluctuation width and the elasticity do not change.

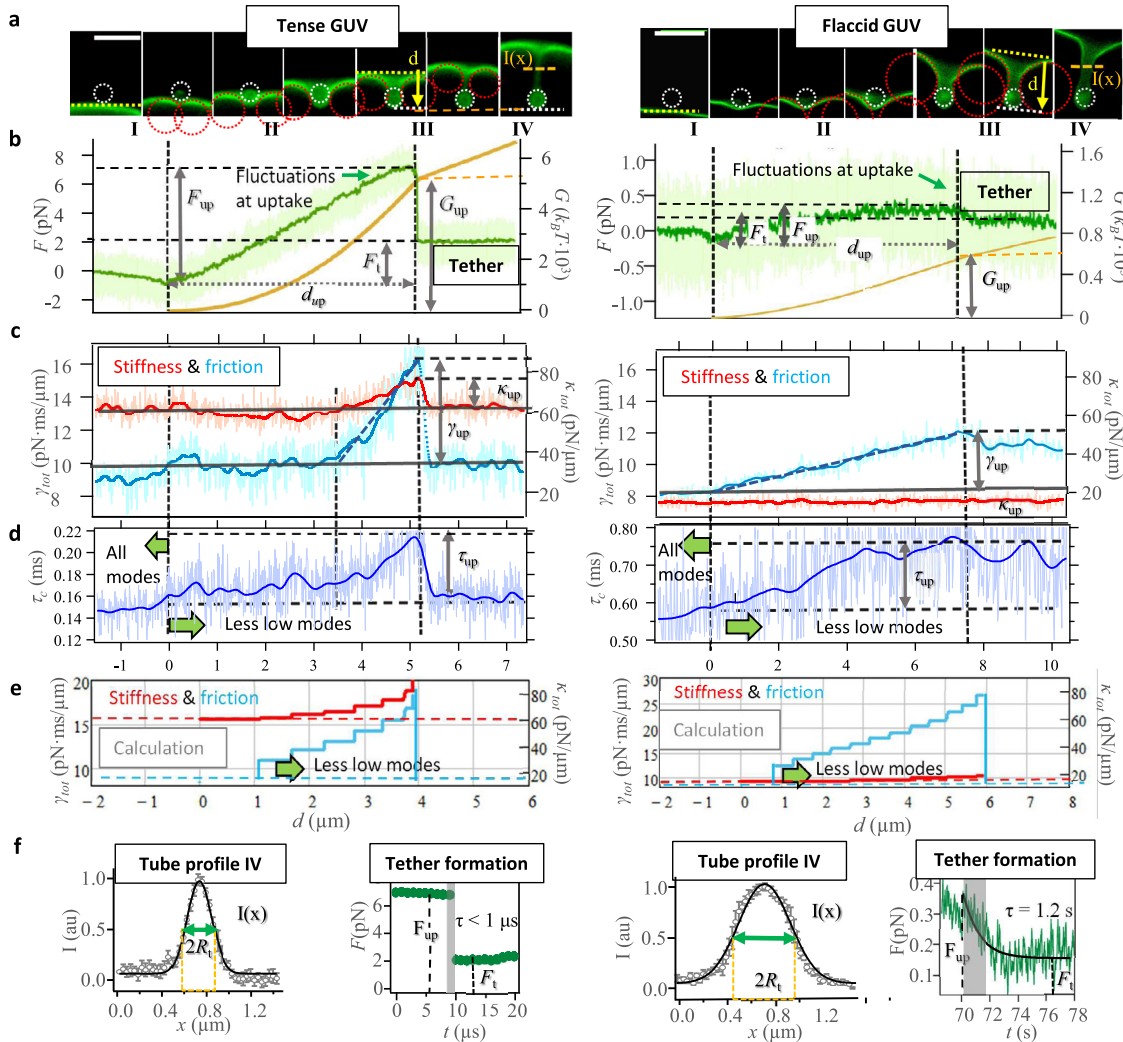

**Fig. 3 | Membrane deformations and fluctuations change during indentation.**
**a** Series of fluorescent images (scale bars: 3 μm) with circles indicating the bead (white) and the torus (red). Four stages are identified: I. Before contact at $d < 0$, II. Membrane deformation and wrapping for $d < d_{up}$, III. Uptake and membrane tube formation at $d \geq d_{up}$, and IV. Membrane tube elongation at constant force.
**b** Deformation force $F(d)$ (left axis, green) and deformation energy $G(d)$ (right axis, orange) profiles versus indentation distance during induced uptake of a tense (left column) and flaccid (right column) DOPC vesicle. The force fluctuation during the uptake corresponds to the bright data points and the mean value of the force smoothed over $\Delta t = 100$ ms corresponds to the dark curves. $F_{up}$ and $G_{up}$ indicate the uptake force and uptake energy, respectively, occurring at the distance $d_{up}$. After uptake, a membrane tube is elongated by a force $F_t$. **c** Particle position fluctuation changes expressed by the stiffness $\kappa_{tot} = \kappa_{opt} + \kappa_{mem}(d)$ (right axis, red) and the friction coefficient $\gamma_{tot} = \gamma_{bd} + \gamma_{mem}(d)$ (left axis, light blue). The fluctuations in $\kappa_{tot}$

and $\gamma_{tot}$ during the uptake correspond to the bright lines and their corresponding mean values smoothed over $\Delta t = 100$ ms are represented by the dark solid curves. $\kappa_{up}$ and $\gamma_{up}$ are the uptake values for the stiffness and friction coefficient at $d_{up}$. **d** Distance-dependent increase in autocorrelation time $\tau_c(d)$ for tense and flaccid vesicle. The fluctuations in autocorrelation time $\tau_c$ during uptake are shown in bright blue, their mean values smoothed over $\Delta t = 100$ ms in dark blue with $\tau_{up} = \tau_c(d_{up})$. **e** Calculated increase in friction and stiffness by subsequent suppression of lower modes with distance $d$. **f** Tube profiles (left): lateral fluorescence mean intensity $I(x)$ measured from images at stage IV at different positions across the tube (gray open circles). Error bars correspond to the standard deviation of the mean. A Gaussian fit (black solid line) is used to obtain the tether radius $R_t$. Tether formation (right): a close-up of the transition stage III with a sudden decay (tense GUV) and a smooth decay (flaccid GUV) of the force from $F_{up}$ to $F_t$ described by the transition time $\tau$.

**Table 1 | Friction and stiffness parameters at room temperature for a 1 μm sized bead in aqueous solution depending on the time-averaged penetration distance $d$ to the GUV membrane**

| Bead friction $\gamma_{bd}$ | Membrane friction $\gamma_{mem}(d)$ | Trap stiffness $\kappa_{opt}$ | Membrane stiffness $\kappa_{mem}(d)$ | Relaxation times bead $\tau_{tot}(d)$ | Relaxation time[a], mode $\tau(n = 1-10)$ | Trap dwell time $\Delta t$ | Membrane bending rigidity $K$ | Membrane surface tension $\sigma$ |
|---|---|---|---|---|---|---|---|---|
| $8 \frac{pN}{\mu m}$ ms | $0-8 \frac{pN}{\mu m}$ ms | $4-75 \frac{pN}{\mu m}$ | $0-15 \frac{pN}{\mu m}$ | 0.1–0.8 ms | 56–0.19 ms | 100 ms | $1.6-20 k_B T$ | $10-300 \frac{k_B T}{\mu m^2}$ |

The surface tension converts as $250\, k_B T/\mu m^2 = 250 \cdot (4 pN \cdot nm/\mu m^2) = 1\, pN/\mu m$. Shortest particle relaxation times inside the trap are $\tau_c = \gamma_{tot}/\kappa_{opt} \approx 0.1 - 2$ ms (lateral $xy$ directions). 100,000 3D particle positions were recorded at each trap position, i.e. within $\Delta t = 100$ ms. [a]Theoretical values for a GUV with $R_g = 10\, \mu m$, $K = 10\, k_B T$, $\sigma = 50\, k_B T/\mu m^2$.

The uptake distance for the flaccid vesicle $d_{up} = 7\,\mu m$ is larger than $d_{up}$ of the tense GUV, probably due to the membrane excess in flaccid vesicles pulled out to produce the internalization of the particle. The friction experienced by the particle partially wrapped into the flaccid vesicle and

the resulting relaxation time $\tau_c(d)$ increase continuously until uptake occurring at stage III (see Supplementary Movies 3, 4).

The calculation results in Fig. 3e, based on the theory introduced above reveal the same behavior: the fluctuation amplitude is limited by

**a** Uptake energy decreases with flaccidness

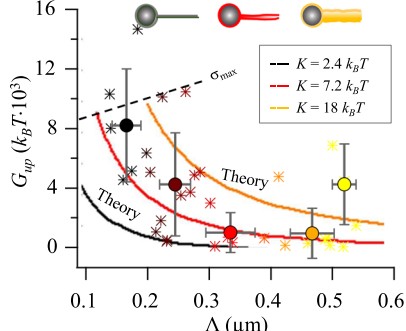

**b** Uptake force decreases with flaccidness

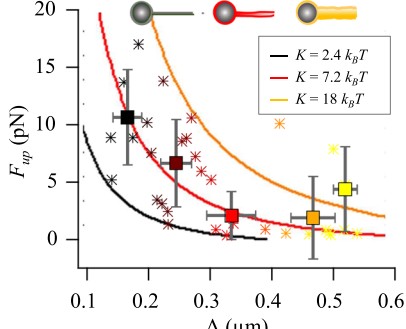

**c** Uptake length changes with flaccidness

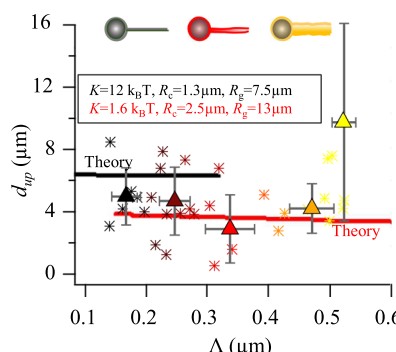

**d** Uptake profile changes with flaccidness

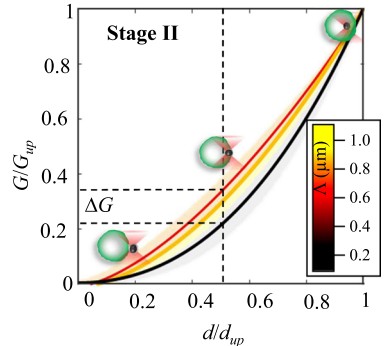

**Fig. 4 | Energetic cost of particle uptake in DOPC-GUVs. a** Uptake energy $G_{up}(\Lambda)$ in $10^3 \cdot k_B T$ and **b** uptake force $F_{up}(\Lambda)$ in $pN$ decrease with the characteristic membrane length $\Lambda$, i.e. with increasing flaccidness of the GUVs (see increasing tube radius and color change from black to yellow). Theoretical curves $G_{up}(\sigma)$ and $F_{up}(\sigma)$ are plotted against $\Lambda(\sigma)$ and are shown for bending rigidities $K = 2.4, 7.2$ and $18\,k_B T$ in black, red and orange. The black dashed lines indicate $\sigma_{max} = 400\,k_B T/\mu m^2$ and $\sigma_{mim} = 15\,k_B T/\mu m^2$. **c** Dependency of uptake length $d_{up}(\Lambda)$ on length $\Lambda$. For flaccid vesicles ($\Lambda > 0.5\,\mu m$) $d_{up}$ increases. The calculated relationship $d_{up}(\Lambda)$ (or $d_{up}(\sigma)$ versus $\Lambda(\sigma)$) is shown for two known GUV radii $R_g$ and

torus radii $R_c$ (black and red solid lines). **d** Normalized energy profiles $G(d/d_{up})/G_{up}$ presented as the mean of three different groups $0.1 < \Lambda_1 \leq 0.2$, $0.2 < \Lambda_2 \leq 0.4$ and $0.4 < \Lambda_3 \leq 0.6$ (solid lines). Shaded regions represent standard deviations of the mean of all experiments with complete uptake process. $\Delta G$ indicates a difference in energy invested on halfway to uptake ($d/d_{up} = 0.5$) between tense and flaccid vesicles. For figures **a** to **c:** single data points (asterisks) correspond to single experiments and pooled data (symbols: circles for $G_{up}$, squares for $F_{up}$ and triangles for $d_{up}$) represent the means of the corresponding variables for intervals of $\Delta\Lambda = 0.1$. Error bars represent the standard deviation of the mean in each case.

the optical trap leading to a constant stiffness for both vesicles (red curves). The friction of the particle-membrane system increases linearly (light blue curves) based on the stepwise suppression of lower modes ($n_0$ increases, $N_{mx} = R_g/R_b = 15$ for the tense and $N_{mx} = 26$ for the flaccid GUV) as described in Eq. (7). However, to achieve friction values comparable to the experiments, not all modes up to $N_{mx}$ over the distance $d_{up}$ were suppressed, but over the distance $2 \cdot d_{up}$ in the denominator in Eq. (7). The mode suppression rate with $d$ needs to be investigated in more detail in the future.

After uptake, the parameters $\gamma_{tot}(d > d_{up})$ and $\kappa_{tot}(d > d_{up})$, describing the fluctuation amplitude and relaxation, drop off significantly when the particle is connected only to the membrane tether.

**The membrane fluctuations set the energy costs for particle uptake**

The energy costs of the particle uptake process depend strongly on the membrane mechanical state as can be seen by comparing the energy curves $G(d)$ of both vesicles (Fig. 3b). In total, 32 experiments with bead uptake into DOPC lipid GUVs with different membrane properties were analyzed. To avoid the possibility of undesired lipids attaching to the bead due to repeated experiments on the same vesicle, the data presented in this figure were obtained from $N = 32$ independent GUVs indented only once. We varied the osmotic difference between the GUVs interior and the external buffer (see "Methods" section) resulting in different GUVs (with $R_g$ ranging from 7 to 15.6 $\mu m$, with bending rigidity $K$ ranging from $1.6 k_B T - 38 k_B T$ and with surface tension $\sigma$ ranging from

$4.9\,k_B T/\mu m^2 - 300 k_B T/\mu m^2$. Because the pronounced difference in their membrane properties, we sorted the vesicles in different groups according to their characteristic length $\Lambda$.

In Fig. 4 we present the main parameters that characterized the uptake process—colored and plotted as a function of $\Lambda$ (color bar in Fig. 4d). Figure 4a shows the measured uptake energies $G_{up}$ for the DOPC-GUVs (asterisks for single data) and their corresponding pooled data points over intervals of $\Delta\Lambda = 0.1\,\mu m$ (filled circles). The error bars along the $x$- and $y$-axis correspond to the standard deviation of the mean from the pooled data. One can see that highest uptake energies (dark markers) are required for tense vesicles ($\Lambda \approx 0.15\,\mu m$) with an average energy value of $8.2 \pm 3.8 \cdot 1000\,k_B T$. The decrease in the membrane tension contribution with respect to the bending energy (reddish markers, $\Lambda \approx 0.3\,\mu m$) turns the uptake process more favorable by lowering $G_{up}$ about 8 times on average. Nevertheless, as the bending contribution dominates over tension and the membrane fluctuations become more evident (yellowish markers, $\Lambda \approx 0.5\,\mu m$), the uptake costs $G_{up}$ tend to rise again, thus making the particle uptake less favorable for a fluctuating membrane.

Figure 4b, c displays how the uptake force $F_{up}$ and the uptake distance $d_{up}$ vary over $\Lambda$, respectively. Less uptake force $F_{up}$ is required for more flaccid vesicles, i.e. with increasing $\Lambda$—similar to the behavior of $G_{up}(\Lambda)$. The uptake distance $d_{up}(\Lambda)$ is shorter for intermediate values of $\Lambda$, but increases again for more flaccid vesicles exhibiting membrane excess.

The nonlinear decay of the uptake costs, expressed by $G_{up}$ and $F_{up}$, can be also confirmed by the mathematical model. In overlay to the experimental data, three calculated curves for different bending

**a** Membrane-particle coatings

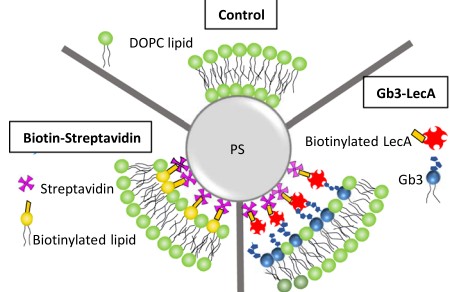

**b** Uptake costs increase with stiffness

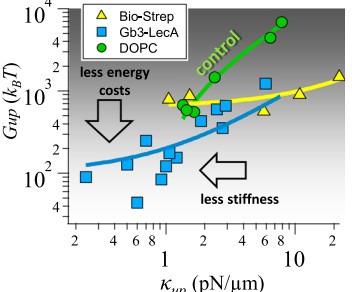

**c** Uptake costs increase with friction

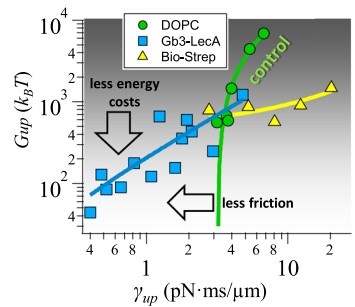

**d** Friction correlates with stiffness

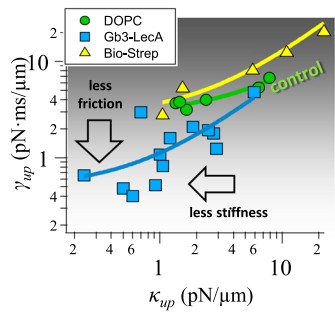

**Fig. 5 | Adhesion-mediated particle uptake into functionalized GUVs.**
**a** Overview sketching the different membrane-particle coating conditions: the DOPC membrane/Polystyrene particle (PS) is used as control (DOPC markers and linear fits in green, $N = 6$ independent experiments), biotinylated−DOPC/Strepta-vidin-coated particles (Bio-Step markers and linear fits in yellow, $N = 5$ independent experiments), and glycosphingolipid Gb3-DOPC/LecA-coated particles (Gb3-LecA markers and linear fits in blue, $N = 13$ independent experiments). Comparison of **b** uptake energies $G_{up}(\kappa_{up})$ increasing with the stiffness at uptake $\kappa_{up}$ and **c** uptake energies $G_{up}(\gamma_{up})$ increasing with friction at uptake $\gamma_{up}$. **d** Approximately linear relation between uptake friction and uptake stiffnesses. The solid curves in b, c, and d represent linear fits in double logarithmic plots.

rigidities $K$ reproduce the measured results reasonably, where $\Lambda = \sqrt{K/\sigma}$ was varied by $\sigma$. The black dashed lines limiting the theoretical curves indicate the maximum tension $\sigma_{max} = 400\,k_BT/\mu m^2$ used (see also Supplementary Fig. S19).

Now, in order to study how the indentation energy $G(d)$ changes during the uptake process we compare the normalized energy profiles of DOPC-GUVs sorted according to $\Lambda$. We normalize the energy profiles $G$ (see Fig. 3b) by $G_{up}$, the energy is summoned up directly before particle internalization. Likewise, the distance axis was normalized by the uptake distance $d_{up}$. For visualization purposes, we sort the normalized energy profiles $G(d/d_{up})/G_{up}$ in three different groups: $0.1 < \Lambda_1 \leq 0.2$, $0.2 < \Lambda_2 \leq 0.4$, and $0.4 < \Lambda_3 \leq 0.6$. Figure 4d displays the average indentation profiles of each group (solid lines) and the standard deviation of their corresponding means (shadow region). It can be seen that the energy profiles differ in shape, not depending on uptake length and GUV size, but only on the membrane properties expressed by $\Lambda$. Remarkably, the graph reveals that the deformation of more flaccid vesicles (red curves) requires about 50% more energy over the first half of indentation than tenser vesicles (black colors)[10,21].

**Particle coating and membrane composition determine fluctuations and uptake costs**
We have characterized the uptake process of uncoated (carboxylated) PS particles into DOPC lipid membranes−without any specific adhesion contribution enhancing the particle wrapping. In this section, we explore the effect of adhesion energies mediated by specific linkers on the uptake process[22,23] and the energy costs $G_{up}$, which was provided by adhesion and by optical forces. However, instead of simply comparing uptake energies, we investigated the relation between uptake energy costs and the bead parameters $\kappa_{up}$ and $\gamma_{up}$, probing the thermal membrane fluctuations as demonstrated in Fig. 5.

We compared three different membrane-particle systems as sketched in Fig. 5a. First, we used a biotinylated membrane and a streptavidin-coated particle. Second, inspired by the lipid-zipper mechanism used by the bacterium *Pseudomonas aeruginosa* to invade host cells and Gb3-containing vesicles[5], we explored the role of the Gb3-LecA interaction on the particle uptake process. Here, glyco-sphingolipid Gb3 was inserted in the DOPC membrane to interact with a particle coated with the bacterial lectin LecA (see Fig. 5a and "Methods" section). Third, for a useful comparison, we use DOPC vesicles interacting with uncoated PS particles as control group.

Figure 5b, c shows in double logarithmic scale the uptake energies $G_{up}(\kappa_{up})$ varying with the maximum membrane stiffness $\kappa_{up}$ and friction $\gamma_{up}$ measured directly before uptake. The solid curves are linear fits to the data points and indicate an approximately linear relation between uptake energy costs and membrane stiffness, or membrane friction, respectively. Both dependencies are most pronounced (largest $\frac{\partial}{\partial\kappa_{up}}G_{up}(\kappa_{up})$ and $\frac{\partial}{\partial\gamma_{up}}G_{up}(\gamma_{up})$) for the control case of pure DOPC lipids (in green), which reveals the highest uptake energies ($G_{up} \approx 700 - 7000\,k_BT$). These energy costs occur for stiffnesses $\kappa_{up} \approx 1 - 8\,pN/\mu m$ ($1\,pN/\mu m = 250\,k_BT/\mu m^2$) and for a limited increase in friction ($\gamma_{up} \approx 3 - 6\,pN \cdot ms/\mu m$).

The biotin-streptavidin system (in yellow), on the other hand, varies only slightly in energy ($G_{up} \approx 600 - 1100\,k_BT$) relative to the control, despite the larger range of stiffnesses ($\kappa_{up} \approx 1 - 21pN/\mu m$) and friction values ($\gamma_{up} \approx 3 - 20pN \cdot ms/\mu m$).

A completely different behavior is shown for the interaction between LecA-coated particles and Gb3-enriched membranes (in blue). Uptake energies were significantly lowered ($G_{up} \approx 300 - 1000\,k_BT$), on average 8-fold relative to the DOPC control case. Likewise, stiffness and friction were also decreased by up to an order of magnitude: $\kappa_{up} \approx 0.2 - 6\,pN/\mu m$ and $\gamma_{up} \approx 0.4 - 5\,pN\,ms/\mu m$.

The additional adhesion energy by specific interactions reduces the uptake energy directly. The membrane composition influences the stiffness and friction experienced by a particle wrapped into it and thereby controls the deformation and uptake energy. This is summarized in Fig. 5d, where the maximal friction and maximal stiffness right before particle uptake are plotted against each other. Linear fits indicate an approximately linear increase of friction with stiffness. This increase is relatively small for the DOPC control case ($\frac{\partial}{\partial \kappa_{up}} \gamma_{up}\left(\kappa_{up}\right)$ small), but is significantly more pronounced for biotin-streptavidin and for LecA-Gb3. Figure 5d also summarizes the high frictions and stiffnesses associated to biotin-streptavidin system relative to the roughly 10-fold smaller membrane frictions and stiffnesses associated with LecA-Gb3.

## Discussion

Understanding the mechanisms how particles like viruses, bacteria or particulates bind to cell membranes and their possible uptake into the cell is of great importance to decipher and fight infections.

We have studied the process of particle binding and uptake into minimized cells (GUVs) using optical tweezers. With our photonic force microscope, we could measure not only mean interaction forces between particle and membrane, but also fluctuations of forces and particle positions at the very high tracking rate of 1 MHz. This temporal resolution allowed us to extract particle fluctuation parameters such as the stiffness $\kappa_{tot}(d)$ and the friction $\gamma_{opt}(d)$ experienced by the particle interacting with the membrane. Since we could determine these physical parameters at each particle position $d$ and at each membrane deformation, we gained significantly more information than by pure analysis of their mean values. However, the interpretation of changes in stiffness and friction was difficult and first explanation attempts were counterintuitive. Therefore, we developed a mathematical model to be able to explain all our observations and thereby to gain useful interpretations of the measured phenomena. In this way, we could understand the significantly different interactions of functionalized particles and tailored membrane compositions, which play an essential role for bacterial infection mechanisms.

In the following sections we want to discuss and assess our approaches and results.

### The experimental approach

Photonic force microscopes are optical tweezers-based systems that use the intrinsic thermal motion of a trapped probe as an additional scan mechanism to the movable optical trap. By nanometer precise 3D tracking at 1 MHz, we were able to measure particle motions on very short timescales (typically $\tau_0 < 30\,\mu s$), on which the particles diffuse almost freely. This means the particles do not see potentials having curvatures of $\kappa > \gamma/(10 \cdot \tau_0)$ corresponding to potential autocorrelation times of $\tau_c \approx 10 \cdot \tau_0 = 300\,\mu s$. Assuming nearly thermal equilibrium and accepting a 10% error, short-time free particle diffusion allows to extract the friction coefficient $\gamma = k_B T/D$. The diffusion coefficient $D$ is obtained from the linear part of the exponentially decaying autocorrelation function as explained in this study. This concept and related approaches are known for a long time[24] and have been tested and applied by us in many different situations[10,11,25,26]. Figure 1d confirms the highly linearly decay of the autocorrelation curves $AC\left[b(t)\right] = AC(\tau)$ for $t < 30\,\mu s$ for particles that barely contact the membrane as well as for particles fully wrapped into it.

A second, often questioned aspect is the robustness of particle tracking in close proximity of scatterers such as other particles or cells. However, these affect mainly the mean positions and forces by typically less than 20% for $1\,\mu m$ trapped particles, but do not affect the particle position fluctuations defining the stiffness $\kappa = k_B T/\langle b^2 \rangle$ and the friction coefficient $\gamma$[10,11].

Another important question is whether a $1\,\mu m$ particle is appropriate to explore the membrane fluctuation properties. If one is interested in mere membrane fluctuations then approaches such as focusing a laser beam onto a GUV membrane are useful and sufficient[9,27]. Other approaches use optically trapped particles as probes to transfer the membrane fluctuations onto the particle[13,28]. Related measurement techniques exploit angular motion tracking of the particle in contact with cell membranes to estimate stiffnesses and angular frictions through power spectral density analyses up to 4000 Hz[29]. However, when particle-membrane interactions or particle-induced changes in local friction and stiffness are of interest (possibly to facilitate particle entry), extended approaches to understand and analyze fluctuation data must be applied, which we pursued in this paper.

### Can higher mode fluctuations make a membrane stiffer and more viscous?

Such a nonintuitive question needs to be discussed and understood, which was the motivation for the theory part of this study. The decisive starting point is to assume that the membrane forces and optical forces are connected in parallel, while different membrane modes (corresponding to springs with different elasticities and frictions) are connected in series to each other. In such configurations, the fluctuation amplitudes add up independently and each fluctuation mode relaxes independently into its own thermal equilibrium with the same free thermal energy $k_B T$. To determine the total friction and the total stiffness of the system, we derived the response function $\alpha_{tot}(\omega, N)$ from the Langevin equation in temporal and spatial Fourier space. For mathematical convenience, we assumed the same white noise for the particle and all membrane modes $n < N$. By calculating the zero frequency limit $\alpha_{tot}(\omega \to 0, N)$ we could determine the total stiffness $\kappa_{opt} + \kappa_{mem}(N)$ experienced by the particle, from the high-frequency limit $\alpha_{tot}(\omega \to \infty, N)$ we could determine the total friction $\gamma_{bd} + \gamma_{mem}(N)$.

The total stiffness and the total friction could be determined experimentally from the short-time autocorrelation analysis. The membrane stiffness $\kappa_{mem}(N)$ and friction $\gamma_{mem}(N)$ are obtained by the inverse summations of each $\kappa_{mq}$ and each $\gamma_{mq}$ from the individual fluctuation modes $n < N$. Each mode is a function of the membrane's surface tension $\sigma$ and bending modulus $K$, such that the measured fluctuation parameters $\kappa_{mem}$ and $\gamma_{mem}$ can be transferred into membrane parameters $K$ and $\sigma$.

By the linearized autocorrelation functions, we could add up the autocorrelation function of each mode to obtain the total autocorrelation function containing the total relaxation time. In this way we could cross-check the unintuitive result that the relaxation time decreases with a higher mode sum $N$. Through the separate analysis of $\kappa_{mem}(N)$ and $\gamma_{mem}(N)$, we found that the shorter fluctuation relaxation $\tau_{tot}(N) = \gamma_{tot}(N)/\kappa_{tot}(N)$ is mainly based on the total viscous drag, which reduces with higher $N$. After having established and tested this theoretical approach, it was possible to investigate how the total relaxation time or viscosity changes when the smallest mode $n_O$ is increased instead of the highest mode $N$ in the mode sum $\sum_{n_0}^{N}(\ldots)$. This corresponds to the effect that low $n$ modes, i.e. long wavelength modes are subsequently suppressed. Exactly this suppression effect leads now to an increase in relaxation time or friction, as shown by the solid lines in Fig. 2c. This initial and totally unexpected increase in friction was also visible in the experiments, when the particle was slowly indented and wrapped into the membrane of the GUV (see Fig. 3). As visible in Fig. 3b, c the amplitude of the fluctuations hardly changes with $d$, i.e. the stiffness $\kappa_{mem}$ remains nearly constant. Even in the case for the tense vesicle $\kappa_{mem}$ increases by only 20% (60 pN/$\mu m$ to 74 pN/$\mu m$), indicating that $\gamma_{mem}(N)$ controls the temporal fluctuation behavior and is the dominant parameter during uptake.

A suppression of lower frequency modes can also be perceived when the strings of a guitar or a cello are subjected to higher tension by pressing/pulling with a finger in the middle of the string. The higher

the tension, the higher becomes the tone because of the suppression of lower oscillation modes.

The second part of the theory section was a reapplication of a calculation approach published earlier[10]. In that work we assumed that the global membrane deformation $h_{gl}$ is always proportional to the local membrane deformation $h_{lo}$. In the present study (see Eq. (8) and Supplementary Note 5), we ensured that the local and global membrane deformation forces were the same at each position (serial connection). Although the correct total deformation $d_{up}$ is underestimated, the theoretically predicted uptake energies and uptake forces profiles are very similar to those obtained in the experiments when the known membrane parameters bending rigidity $K$ and tension $\sigma$, as well as the trap stiffness $\kappa_{opt}$, were fed into the calculations (see Fig. 2f).

The most robust and reproducible values were measured briefly before uptake, i.e., at position $d < d_{up}$. For $d > d_{up}$ we determined the sudden drop in energy and force, $G_{up}$ and $F_{up}$, as well as the drop in friction and stiffness, $\gamma_{up}$ and $\kappa_{up}$.

### Particle uptake into tense vesicles costs more energy

While the tether forces $F_t$ after particle uptake into the GUV could be determined precisely with our PFM, the measurement of thin tube diameters, especially for tense vesicles, leads to 20% error by fluorescence imaging with a NA = 1.2 objective lens. The same 20% uncertainty results for the estimation of the membrane parameters $K$, $\sigma$ and $\Lambda = \sqrt{K/\sigma}$, which does not affect any of our results and conclusions.

As demonstrated in Fig. 4, we sorted 32 DOPC lipid vesicles according to their mechanical membrane properties bending rigidity $K$ and tension $\sigma$, i.e. into groups (pooled data) with different degrees of flaccidness (expressed by $\Lambda = \sqrt{K/\sigma}$). Interestingly, it turned out that the uptake energies and forces decrease in a characteristic and reproducible manner with the parameter $\Lambda$. In addition, we could reproduce this relationship also by the mathematical model using the experimental parameters extracted as input.

While the uptake length $d_{up}$ for differently tense vesicles varies by about less than 50% (Fig. 4), $d_{up}$ increases significantly for very flaccid vesicles ($\Lambda > 0.5\,\mu m$), which can be explained by the available membrane excess of the GUVs, necessary to allow fluctuations with pronounced amplitudes.

We show typical uptake profiles in Fig. 4 by normalizing the profiles in energy and indentation distance. We observe that for flaccid GUVs, almost 40% of their required $G_{up}$ is readily invested on halfway to uptake, whereas tense GUVs required only 20% (dashed lines), such that nearly 80% of the uptake energy is required for indentations $d > 0.5 \cdot d_{up}$. A similar effect is apparent in Fig. 3c for $\kappa_{tot}(d > 3.5\,\mu m)$ and $\gamma_{tot}(d > 3.5\,\mu m)$, i.e. after halfway to uptake. The smooth increase for flaccid vesicles might be an effect of membrane excess and large undulations. Differences on the courses followed by the energy profiles are attributed to the global and local deformations that vesicles undergo during the indentation process[10].

### Strong adhesion energies suppress membrane fluctuations, while LecA−Gb3 interactions increase them

How can the energy costs for particle uptake be reduced and the uptake probability increased? Are energy costs affected by the particle fluctuations or can particle fluctuations predict the energy costs? These questions were addressed by the experiments summarized in Fig. 5, where we used different particle coatings and membrane compositions to induce particle uptakes in vesicles with similar sizes. DOPC lipid membranes with uncoated particles were taken as control condition, revealing an approximately linear increase of uptake energy $G_{up}$ with uptake friction $\gamma_{up}$ and uptake stiffness $\kappa_{up}$. Typically, relatively high energy costs ($700 − 7000\,k_BT$) were associated with high stiffnesses and frictions as indicated by the green curves in Fig. 5b, c. When we indented streptavidin-coated beads into biotinylated lipid membranes (yellow curves), the uptake

energies were lower and did not vary much over a broad range of measured frictions and stiffnesses. However, the strong biotin-avidin connection enabled higher values for friction and stiffness than those in the DOPC control system. Possibly, a distance-dependent stable patch between the vesicle and the particle is formed upon contact. As the contact area increases with distance, more bonds are formed restricting the bead fluctuation amplitude, as well as slipping and membrane kinks against the bead[30].

As a system interesting to infection biology, we investigated the interaction between DOPC lipid membrane enriched with the glyco-sphingolipid Gb3 (as receptor) and the bacterial lectin LecA, which was coated on the beads. In this case not only the uptake energies were lowered by an order of magnitude (blue curves), also the friction factors and stiffnesses were significantly reduced, which was also displayed in Fig. 5d. This means that the membrane fluctuations were larger in amplitude (lower stiffness) with shorter relaxation times (lower friction) during the whole particle indentation and uptake. It was found on the one hand that LecA-Gb3 interactions enhance molecular adhesion[21], on the other hand, that the LecA-Gb3-induced lipid-zipper mechanism reduces the membrane deformation energy and uptake energy[5]. From this, we conclude that wrapping the particle into the floppy membrane is easier, leading to a faster increasing adhesion area and therefore facilitating particle uptake as shown in Fig. 5. Hence, thermal energy fluctuations of the membrane contribute to the particle uptake, without requiring significant deformation forces from the trap.

It is worth to mention that only in Gb3-vesicles we observed the spontaneous formation of small intra-vesicles and inward protrusions, which is characteristic for the existence of membrane reservoirs in vesicles with negative preferred curvature[31].

In summary, we addressed the question how membrane-bound particle fluctuations on scales of nanometers and microseconds are related to the complete particle uptake into the GUV taking place over several micrometers and tens of seconds. Such membrane fluctuations are well visible under the microscope but are complex in their spatial and temporal behavior. By 1 MHz particle tracking with a photonic force microscope, we were able to extract characteristic short-time fluctuation parameters, i.e. the stiffness and friction experienced by the particle interacting with the membrane. Unexpectedly, both values increased during the indentation of the particle into the GUV until uptake, mainly dominated by friction. The changes in friction and stiffness experienced by the particle could only be understood by developing a mathematical model, which could reproduce all measured indentation and which allowed us to draw important conclusions. By decomposing the membrane fluctuations in spatial and temporal frequencies, we could identify the relation between the biochemical membrane composition and physical fluctuation parameters of differently coated particles. The suppression of low-frequency fluctuation modes with ongoing particle indentation led to an increase especially in membrane friction. Interestingly, we could show that the bacterial lectin LecA binding to its host cell receptor, the glycosphingolipid Gb3, is able to control the local membrane friction, thereby reducing its energy costs and favouring particle entry. Our study points out the tremendous importance of thermal fluctuation measurements and computer models to understand molecular scale interactions.

## Methods
### Samples preparation
**DOPC-GUVs**. The DOPC-GUVs were prepared using lipids mixed in organic solvents following the standard electroformation protocol[32]. Briefly, 99 mol% DOPC (1,2-Dioleoyl-sn-glycero-3-phosphocholine) lipids were mixed with 1 mol% Texas Red-DHPE (1,2-dihexadecanoyl-sn-gly-cero-3-phosphoethanolamine) fluorescent lipid in chloroform. The lipid mixture was deposited onto two ITO-coated glass slides (14 $\mu l$ on each conductive face). After an overnight period to ensure the chloroform evaporation, a closed chamber was formed with the ITO-slides and filled

with 3 M of sucrose solution (Carl Roth). Flaccid vesicles were produced by inducing an osmotic difference of $200 mOsmL^{-1}$ between the inner sucrose solution and the external DPBS buffer solution.

**Gb3-GUVs.** A mixture of 1 mol% of purified porcine glycosphingolipid Gb3 (Matreya, LLC), 98 mol% DOPC, and 1 mol% Texas Red-DHPE was used to electroform Gb3-containing vesicles following the above described protocol.

**Biotin-GUVs.** We used 1 mol% of biotinylated lipids (FSL-biotin Sigma Aldrich), 98 mol% of DOPC and 1 mol% of Texas Red-DHPE. A more detailed description of the protocols related to GUVs can be found in ref. [21].

## Particle Functionalization
For control experiments we used 1 μm sized carboxylated polysterene (latex) microspheres (Polyscience, Inc). For the experiments probing the Biotin-Streptavidin interaction we used 1 μm sized streptavidin-coated microspheres (Polyscience, Inc).

**LecA coating.** For the experiments testing the Gb3-LecA interaction, we used 1 μm sized streptavidin-coated latex microspheres (Polyscience, Inc) to bind biotinylated lectin LecA on their surfaces following the procedure suggested by the supplier (Polyscience, Inc). Briefly, 50 μl of streptavidin-coated microspheres from the original solution (particle concentration 1.25%) were centrifuged and gently washed 3 times using a PBS/BSA binding buffer at pH 7.4 and room temperature. The washed microspheres were resuspended in 1 ml of the binding buffer solution and subsequently incubated with 40 μl of biotinylated LecA solution (concentration of 0.56 mg/ml). This is equivalent to have 33 mg of LecA per mg of latex polymer which was enough to have LecA-coverage on most of the latex microspheres. To verify the presence of LecA on the latex microsphere, we incubated them with streptavidin Alexa Fluor 488 conjugate, which binds to the biotinylated LecA. We observed the fluorescence of the sample under confocal microscope. Although not all of the microspheres presented fluorescence most of them did it, confirming the presence of LecA on the latex surface[21].

The linker densities for Gb3-DOPC/LecA and biotinylated−DOPC/Streptavidin experiments showed in Fig. 5 are approximately 607 μm$^{-2}$ and 450 μm$^{-2}$, respectively. They were obtained from the detachment energy measured from pulling experiments in our previous work[21].

**Characterization of vesicle elastic properties.** The electroformation method used to produce our samples does not guarantee the production of GUVs with the same tension. Since experiments were performed in different GUVs, without repeating the experiment in the same vesicle, it was fundamental to characterize them individually. After a successful uptake event, an inward membrane nanotube or tether is formed. The force drops abruptly to a constant value $F_t$ that drives the elastic extension of the tether. The free energy minimization that describes the tube formation yields to expressions for the membrane tension $\sigma$ and the bending rigidity $K$, both as a function of tube radius $R_t$ and tether force $F_t$: $\sigma = F_t/(4\pi R_t)$ and $K = F_t R_t/(2\pi)$. The value of $F_t$ is readily obtained from the force−distance curve in Fig. 3b. The value of $R_t$ is measured from the fluorescent images by fitting a Gaussian function to the lateral profile of the tube, as shown in Fig. 3f. The parameter is then defined as $\Lambda = \sqrt{K/\sigma} = \sqrt{2} \cdot R_t$ so that larger $\Lambda$ values are associated to flaccid vesicles while smaller $\Lambda$ to tense ones.

## Reporting summary
Further information on research design is available in the Nature Portfolio Reporting Summary linked to this article.

## Data availability
The data generated in this study are provided within the paper. The supplementary files and Source data are provided with this paper.

## Code availability
Analysis codes are available from the corresponding author upon reasonable request.

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

## Acknowledgements

The authors thank Dr. Thorsten Auth, Dr. Christian Fleck and Prof. Lars Pastewka for fruitful discussions on theoretical models and Fabian Rohrbach for generating Supplementary Movie 7. This work was supported by the German Research Foundation (grants RO 3615/6-1, RO 3615/15-1, RO 3615/3-2 and RO 4341/3-1, and Germany's Excellence Strategy (EXC-2189, project ID 390939984), and by the Freiburg Institute for Advanced Studies (FRIAS).

## Author contributions

A.R. and W.R. conceived the project and obtained the funding. Y.A. performed the experiments and analyzed the data. A.R. developed the theoretical model and performed the simulations. R.O. helped with sample preparation. A.R. and Y.A. interpreted the results and wrote the manuscript.

## Funding

## Competing interests

The authors declare no competing interests.
