## [Peer Review File · Nature Communications]

Thermal fluctuations of the lipid membrane determine particle uptake into Giant Unilamellar VesiclesReviewers' Comments:

Reviewer #1:

Remarks to the Author:

In the manuscript of Ayala et al, the authors describe experiments where an optically trapped bead is brought in contact with a GUV. The aim is to simulate a situation found in nature during material uptake by cells. The subject is widely studied in biology, but up to now experiments that can replicate in a clean way the forces of this process from a pure membrane point of view have never been performed. This is partially due to the complexity of the system and the difficulties of the analysis. In this sense the paper is excellent, as it closes an important gap, and provides clean experimental access in addition to convincing theoretical modelling. By autocorrelation of the spatial fluctuations of the beads, mechanical properties of the membrane are obtained, and the force required for particle uptake is measured in a very convincing way.

Overall, the manuscript is very well written. The experiments and analysis are clearly described and well discussed. Unfortunately, there are a few points that should be addressed before the present reviewer can support publication, however I would like to point out that the paper is a very good fit for nature communications. Find below a list of issues that should be addressed.

General

1. All optical forces are assumed to be sufficiently small such that a linear trap stiffness can be used. Is this approximation valid for the large forces required for beads to enter the GUV? If the laser power is increased to get large forces, what is the expected local heating and can the authors exclude phase transitions due to the heating?
2. The paper is mostly addressed to a readership interested in membrane biophysics. In this community, the letter κ is reserved for the bending rigidity; however, the authors use the letter K to denote bending rigidity and κ for a spring stiffness. I highly recommend to use κ for the bending rigidity, and the letter k for spring stiffness. Otherwise this will lead to severe irritation as readers simply expect that the common notation is used.
3. Equations are all numbered (0)
4. Overall, the manuscript is too detailed for a general readership as expected in Nature Communications. Especially the detailed discussion of the mechanics of membrane fluctuations can be shortened. While it is highly appreciated to have a presentation of the mathematical derivation, it would be much better to focus in the main text on the main findings, and move the mathematical details to the SI. The level of detail hinders the transfer of the main message of the paper to the general audience of Nature Communications.
5. The manuscript contains several language mistakes. I recommend critical reading of the manuscript by an external native speaker.

Specific

6. Line 99: As 1064 nm IR light is strongly absorbed by water, and membranes are sensitive to small changes in temperature via for example a phase transition. Can the authors exclude a temperature effect on the mechanics of the membrane? Alternatively, is it possible to use a wavelength with lower absorption (e.g. 808 nm)?
7. Line 132: the friction is derived for a system with infinite size. Since the bead is touching the membrane, the presented equation for friction is simply not correct, and it should be mentioned that it is an approximation. The additional boundary condition changes the hydrodynamics.
8. Line 146: the term "membrane potential" is widely used to indicate the voltage difference over a membrane. Using different terminology avoids confusion.

9. Line 152: with the bead very close to the GUV, how strong is a lens effect caused by the GUV? The sucrose has a higher refractive index, and the laser is penetrating the GUV to a certain extent.
10. Line 152 / Fig 1C: what causes the sudden jump in b_z when the particles touches the membrane, and why is there no response in b_y ?
11. Line 228: isn't it trivial that the mode sum increases with increasing N ?
12. Line 336 & Fig 3F: I would expect the thinnest tethers to be in the order of ~ 100 nm. The profile in the left graph of Fig 3F could therefore correspond to a diffraction-limited system, rather than the actual thickness. Can the authors comment on this? Also, membrane theory simply predicts that the tether has a radius determined by an interplay between bending rigidity and tension as: $r = \sqrt{\kappa / (2\sigma)}$. The data presented suggests that this widely tested relation is wrong. Is there any explanation for this discrepancy (beside the problem of resolution?)
13. Line 350: it is written that "thermal equilibrium was established". Can the authors demonstrate this, e.g. by calculating $\tau = \gamma / \kappa$ and comparing that to the waiting time at each position?
14. Line 437 / Fig 4B: this is a speculative statement, based on (as far as I can see) a single data point in Fig 4B with large error bars.

Reviewer #2:

Remarks to the Author:

The manuscript of Ayala et al. comprehensively investigates the fluctuations of GUV membranes with different physical conditions and how those fluctuations affect the uptake of a fluctuating microscale particle. The authors make use of experimental techniques in conjunction with analytical theories that help explain their findings. The uptake of particles by membranes itself is an important research area, and it is exciting to see how optical tweezers are applied to visualize the process. Generally speaking, the overall significance of this work is high and the topic attracts a broad spectrum of researchers. However, to further improve the quality of the work and make a more substantial impact on the field, I would suggest that the authors carefully address the following issues.

Abstract:

1. In describing thermal fluctuations, the authors should be more specific. For example, in the sentence "... the uptake energy into a GUV becomes predictable since it increases with smaller and shorter thermal fluctuations", are these fluctuations smaller in amplitude and shorter in wavelength?
2. The sentence "the reduced particle uptake energy using protein-ligand interactions LecA-Gb3 or Biotin-Streptavidin result also from stronger membrane fluctuations" does not seem to be correct, as I thought Biotin-Streptavidin interactions result in reduced fluctuations in the Discussion section.

Result:

3. The equations are not correctly numbered. All equations are mislabeled as equation (0).
4. The authors assume a linear superposition for both the stiffness and friction coefficient to add up the effects of bulk fluid and membrane in the presence of an optical trap. While the elastic term (stiffness) is additive, the viscous time (friction coefficient) typically is not. In fact, I would argue that the drag of the bead would depend on its distance from the membrane. Similarly, once the bead directly interacts (in contact or wrapped) with the membrane, the relaxation of the intervening fluid would make the particle-membrane response highly coupled, and the friction contributions non-additive. Can the authors verify this assumption?
5. How is q_{\max} defined in this work compared with the intrinsic smallest fluctuation length scale of a

free membrane? Can the authors justify their choice of the smallest length scale of the particle-membrane system being R_b ? Perhaps the physical conditions of GUVs should be taken into account in making the decision? For a thermally equilibrated membrane, the fluctuation-dissipation relation should be held for all the relevant undulation modes defined based on the smallest membrane length scale. I wondered whether this choice of length scale precludes other fluctuation modes that could impact the theoretical interpretation of the correlation time in this work.

6. Related to my previous point, an overview of the involved characteristic time and length scales (definitions and values) should be provided. As the particle-membrane problem is essentially a fluid dynamics problem, a discussion about how the authors distinguish between "short" and "long" time scales, "high" and "low" fluctuations (in terms of frequency?), as well as "long" and "short" wavelengths would be necessary. How are these scales or modes related to the correlation time? The authors mention that $30 \mu\text{s}$ is short, but a comparison between this number and other relevant time scales (such as the fluid viscous dissipation time and the particle diffusion time) would make a strong point for fluid mechanists? I noticed that these terms are interchangeably used in the manuscript, and it would be good to have a more unified description for these terms to help the readers relate the observations presented in different figures.

7. The overall presentation of the notations can be improved. The authors should try to simplify the notations and make the necessary number of variables minimal. For example, y and h seem to be representing the membrane shape (deformation), but throughout the main text, h is probably used only once.

8. In line 192, should it be Figure 1A instead of Figure 1B?

9. The line colors in Figure 2A-2C should be consistent. In 2A the membrane-only and bead-only curves are in red and black, while in 2B the membrane-only curve is also in black. Perhaps the authors can change the colors for the bead-membrane curves instead of using blue and red.

10. It is not clear to me what the "Excluding, adding modes" blocks mean in Figure 2C. The green dashed block does not seem to correspond to the top green axis.

11. Can the authors more specifically define " γ " in the section "Membrane deformation by a particle" for the adhesion energy G_{ad} ? I would have thought that the wrapped surface area of the particle would be approximately the total surface area ($4\pi R_b^2$) minus the circular area of the tube ($2\pi R_t$), instead of $2\pi R_b \gamma$. This would affect the determination of the uptake distance d_{up} and hence would deserve more explanation.

12. In line 360, should it be "horizontal dotted lines" in Figure 3B?

13. The authors write, "...to achieve reasonable values, not all modes up to N_{mx} over the distance d_{up} were suppressed, but over the distance $2d_{up}$..." What does this mean? Is there a systematic or rational way of determining the decreasing rate of the number of fluctuation modes?

14. How are the three calculated curves determined in Figure 4? Can the authors specify the equation numbers or remind the readers of expressions for those theoretical curves?

15. What are the membrane surface densities and bead coating densities of those functional groups in the section of adhesion-mediated particle uptake? A correlation between those grafting densities and G_{up} , γ_{up} , and κ_{up} should be given. This information would be important for the theoreticians to compare the results in Figure 5.

Discussion:

16. In Figure 2C, it is shown that τ_c decreases if more higher modes (N) are included. However, in Figure 2D, the autocorrelation function with more modes included exhibits a slower decay, suggesting the opposite trend. Can the authors clarify this?

17. Can the authors also explain how an increase in relaxation time corresponds to an increase in friction in Figure 2C (statement made in line 591)? This would seem contradictory to the argument made in Figure 5 that "the membrane fluctuations are large in amplitude (low stiffness) with long relaxation times (low friction)".

Supplementary:

18. The notations in supplementary text 4 (in terms of h) are inconsistent with the main text (in terms of y), which is confusing. Meanwhile, I find that a lot of the sentences here (page 4 of supplementary)

are redundant, as they have been repeated in the main text.

19. I find the presentations of Figures 3-4, 6-8, 10-17 difficult to digest. They seem to include the source code of a specific software called Mathcad. It would be helpful if the authors can provide paragraphs explaining more details of the figures.

Dear reviewers,

thanks a lot for the large amount of time and efforts you have spent on our manuscript. Your comments, questions and constructive criticism helped us to improve the manuscript.

Let me (A.R.) further apologize for our late reply, but during the Corona times many important people of my group left and I got stuck in obligations.

We did our best to answer all points very carefully, we added and changed figures, we shortened and changed the formulas, we improve the supplementary information. Everything is documented by our point-to-point response.

We are resubmitting the modified version of our paper (with and without indicated changes) as pdf along with the supplementary material, the figures, and videos as requested in the Pre-Production review.

Sincerely,

Alexander Rohrbach and Yareni A. Ayala

RESPONSE TO REVIEWER 1:

Reviewer #1 (Remarks to the Author):

In the manuscript of Ayala et al, the authors describe experiments where an optically trapped bead is brought in contact with a GUV. The aim is to simulate a situation found in nature during material uptake by cells. The subject is widely studied in biology, but up to now experiments that can replicate in a clean way the forces of this process from a pure membrane point of view have never been performed. This is partially due to the complexity of the system and the difficulties of the analysis. In this sense the paper is excellent, as it closes an important gap, and provides clean experimental access in addition to convincing theoretical modelling. By autocorrelation of the spatial fluctuations of the beads, mechanical properties of the membrane are obtained, and the force required for particle uptake is measured in a very convincing way.

Overall, the manuscript is very well written. The experiments and analysis are clearly described and well discussed. Unfortunately, there are a few points that should be addressed before the present reviewer can support publication, however I would like to point out that the paper is a very good fit for nature communications. Find below a list of issues that should be addressed.

General

1. All optical forces are assumed to be sufficiently small such that linear trap stiffness can be used. Is this approximation valid for the large forces required for beads to enter the GUV? If the laser power is increased to get large forces, what is the expected local heating and can the authors exclude phase transitions due to the heating?

Reply: The correct approximation of a linear force range is an important question. Yes, this approximation is valid. If you look at the example of the tense vesicle in figure 1c and 3b, you see that the force of $F_{up} = 7$ pN results in a bead displacement of about 110 nm, which is clearly inside the linear force range. For a $1\mu\text{m}$ bead and a $0.8\mu\text{m}$ laser focus width ($NA = 1.2$), the linear range is about 250 nm in both directions. The same is true for the linear detection range.

Fig. 1c Particle trajectories

The largest, measured optical force of $F_{up} = 17$ pN (Fig. 4b) overcomes this linear force limit slightly, since $17\text{pN} / 7\text{pN} \times 110\text{nm} = 267\text{nm}$. Assuming a 10-20% error for this force value, the message of Fig. 4b remains the same.

We think we can exclude phase conditions since the powers used in our experiments varied only from 3mW to 60mW. These are relatively low powers compared to other GUV experiments reported in literature, where different biological processes were investigated with NIR optical tweezers (OT). These have been turned up to powers of

- 0.5 W without reporting any relevant damage on vesicles (Dimova, 2014).
- 0.23 W focusing directly to membranes (without bead) to manipulate and study single lipid domains on ternary GUVs. Here, a phase transition was observed only at powers above 0.47W, which is equivalent to increasing the temperature at the trap in 7°C (Friddin, 2019).
- 0.19 W to directly merge GUVs (Bolognesi, 2018).

In particular, for DOPC-vesicles changes in morphology of the fluid phase were reported after increasing the temperature of the sample to 50°C for several minutes (25min) (Z. V. Leonenko, 2004). In (Català, 2017) a rise of temperature of 1.9 °C/100mW was reported for experimental condition similar to our experiments. Hence, assuming such increases in temperature in the small region of the GUV illuminated by the trap and the duration of our experiments (~100s or less), our laser powers and local temperature increases would not be enough to induce phase transition.

In the Results section of the manuscript we added the following phrase and included the citation (Català, 2017) :

“We used a low power range to avoid phase transitions or damage on the GUVs during the experiments (Català, 2017), which lasted less than 100s.”

2. The paper is mostly addressed to a readership interested in membrane biophysics. In this community, the letter κ is reserved for the bending rigidity; however, the authors use the letter K to denote bending rigidity and κ for a spring stiffness. I highly recommend to use κ for the bending rigidity, and the letter k for spring stiffness. Otherwise this will lead to severe irritation as readers simply expect that the common notation is used.

Reply: You're right, finding the most appropriate and most intuitive letters for physical quantities is difficult, if not impossible when different communities should be addressed. We chose the letter K for the bending rigidity, since we found this letter in several membrane biophysics papers - which does not mean that your perception about using K is wrong. However, because of the above said and since kappa is a standard letter to express spring constants (or trap stiffnesses) and since I've used these letters in my previous papers, we would like to leave the notation as it is. To meet your criticism and avoid irritation, we now mentioned the term “bending rigidity” nearly every time the letter K was used. Furthermore, we corrected ambiguous descriptions of these quantities.

3. Equations are all numbered (0)

Reply: This must have happened during the PDF conversion process on the Nature Communications website. We are totally sorry for this, since this must have turned the reading of the manuscript and the understanding of the theory extremely difficult.

4. Overall, the manuscript is too detailed for a general readership as expected in Nature Communications. Especially the detailed discussion of the mechanics of membrane fluctuations can be shortened. While it is highly appreciated to have a presentation of the mathematical derivation, it would be much better to focus in the main text on the main findings, and move the mathematical details to the SI. The level of detail hinders the transfer of the main message of the paper to the general audience of Nature Communications.

Reply: In principle I (AR) agree with you that the current manuscript is rather demanding for the general readership of Nature Communications. I'm not sure whether you mean the Discussion or the Results description of the mechanics of membrane fluctuations? I assume you mean the theory in the Results section. However, I think that all the relevant variables that are measured should be explained in the main text. These are the energies G , forces F , stiffnesses κ , friction factors γ , relaxation times τ and the indentation lengths d . They all reach a maximum value at uptake with the subscript X_{up} . Furthermore, I find it essential to understand and explain that the indentation into the GUV results from a global and local deformation, which is briefly explained in the section "Membrane deformation by a particle", which you possibly wanted to have removed. Especially this part is already the largest part of the supplementary material.

In the very end, I shortened the theory part by two equations thereby coming quicker to the important relaxation time of the autocorrelation function. The reading should be easier now.

I did not know what part of the theory, I could move to the supplementary material, without significantly decreasing the reading flux, which should be an important aspect of the manuscript. We hope that the many subheadings in the text help to better follow the story and the take-home messages.

5. The manuscript contains several language mistakes. I recommend critical reading of the manuscript by an external native speaker.

Reply: We both went through the whole text again and corrected several language mistakes. Most of the errors were caused by erroneous voice recognition with my dictation system. We could not find a native speaker doing this job.

Specific

6. Line 99: As 1064 nm IR light is strongly absorbed by water, and membranes are sensitive to small changes in temperature via for example a phase transition. Can the

authors exclude a temperature effect on the mechanics of the membrane? Alternatively, is it possible to use a wavelength with lower absorption (e.g. 808 nm)?

Reply: Based on the explanations given by us above to your question 1, we conclude and strongly believe that phase transitions due to heating and laser radiation do not play a significant role. For several reasons it would be nice to have a highly stable, single mode laser at 808 nm, but we do not have it. A shorter wavelength leads to less absorption, higher intensity gradients and hence offers the same optical forces at less laser power. The absorption in water at 808 nm is about 7 times less than at 1064 nm.

The changes in temperature imposed by the 1064nm laser at the conditions we used are not as large as those reported to induce phase transitions on DOPC-based systems. Any change in GUV mechanical state would be reflected on the mechanical properties measured in the parameter Λ . It is worth to mention that we use the parameter Λ aiming to sort our data to reduce the influence of mechanical parameters of membrane in the main results. Providing accurate values of mechanical properties was not a main goal of our work.

7. Line 132: the friction is derived for a system with infinite size. Since the bead is touching the membrane, the presented equation for friction is simply not correct, and it should be mentioned that it is an approximation. The additional boundary condition changes the hydrodynamics.

Reply: We added a remark about this approximation in the first paragraph of section “Thermal particle fluctuation for different indentation depths”.

8. Line 146: the term “membrane potential” is widely used to indicate the voltage difference over a membrane. Using different terminology avoids confusion.

Reply: We changed it to membrane free energy.

9. Line 152: with the bead very close to the GUV, how strong is a lens effect caused by the GUV? The sucrose has a higher refractive index, and the laser is penetrating the GUV to a certain extend.

Reply: This effect is negligible. This subject was discussed in previous work published by us (Andreas Meinel, 2014). Here we investigated the influence of the light scattered by a GUV on the QPD signal of a trapped 1 μ m polystyrene bead for different laser powers, see figure below extracted from (Andreas Meinel, 2014): The empty trap was scanned at the edge of a GUV for different powers. It was found that on the one hand the QPD detector signal from the GUV alone (S_{\max} and S_{\min}) was small relative to the bead signal (Fig. 3b below); on the other hand, we found that the GUV membrane did not alter or shift the shape or position of the signal $S_g(y)$ along y-direction (S_y in the present work) (Fig. 3c below). From this we concluded that the influence of the GUV, as an additional scatter, on the BFP tracking of the particle was negligible.

Fig. 3 Influence of the GUV on the BFP tracking of the particle. (a) A sweeping laser focus determines the scattering signal from the GUV. (b) Interferometric detector responses $S_g(y)$ from the edges of the GUV obtained by a complete focus scan of length $L_t = 28 \mu\text{m}$. (c) Detector responses $S_g(y)$ from the right edge of the GUV membrane for different laser powers. The signals $S_g(y)$ do not shift laterally. (d) Signal $S_g(y)$ obtained from wave optical simulation.

We mentioned this aspect already in the second paragraph of the discussion section “The experimental approach”.

10. Line 152 / Fig 1C: what causes the sudden jump in b_z when the particles touches the membrane, and why is there no response in b_y ?

Reply: The sudden shifts upwards in position b_z (blue) and downwards in b_y (green) at $t=68\text{s}$ are associated to the moment of uptake completion and tether formation. The contact between membrane and particle occurred at $t=15\text{s}$.

After contact ($t > 15\text{s}$) the GUV membrane shifts the particle continuously out of the trap center by b_y and b_z until the membrane pressure onto the bead is released at $t=68\text{s}$. Here, the bead jumps abruptly back to the trap center. Since the optical trap stiffness is more than five times softer in z -direction than in lateral xy direction, particle displacements in directions of weak counteracting forces (z -direction) occur often. For the same reason, weak forces along x direction are not detected. Hence, our analysis is focused only along the direction in which the uptake experiments were performed (y -direction).

11. Line 228: isn't it trivial that the mode sum increases with increasing N ?

Reply: I assume you are referring to the fluctuation width. You may be right that this statement is trivial for an expert, but I think that many people haven't ever seen the increase in total fluctuation width or in temporal decay of several fluctuating membrane modes. I also think the fact that membrane modes oscillate independently of each other in time and space is not obvious to many non-experts.

12. Line 336 & Fig 3F: I would expect the thinnest tethers to be in the order of ~ 100 nm. The profile in the left graph of Fig 3F could therefore correspond to a diffraction-limited system, rather than the actual thickness. Can the authors comment on this?

Reply: You are right that the tether images have reached the diffraction limit and in terms of optics the cylindrical object could be well below the full width half maximum of the pointspread function (PSF).

However, the following comment of how to estimate the tube diameter should be valid:

Thinner tethers of around 100nm in diameter have been reported in literature to be formed in pulling experiments from high tense vesicles. Usually, micropipettes are used to increase the tension and hold the vesicles (Debjit Roy, 2020) (Renner M, 2011). In our experiments we did not use micropipettes to impose high tension on vesicles. Unlike pulling tubes out of membranes, in our uptake experiments, the stiffest GUVs ($\sigma_{max} = 300 k_B T / \mu m^2$) had membrane tension approximately 10 times less than those held with micropipettes used for pulling experiments.

It is worth mentioning that in some of the high tense vesicles we tested, after a successful uptake, we indeed observed the formation of tethers thinner than the one shown in figure 3f of the manuscript. See the example in figure b below. Unfortunately, these thin tethers could not be measured optically due to low fluorescence and due to the diffraction limit. Hence, such data were not included in our results since we could not sort them by the Λ parameter.

If we had tethers with 100 nm in diameter, they would be accumulated in the darkest regions of figure 4a ($\Lambda < 0.15 \mu m$), since thinner tethers are produced in tense vesicles, which require higher values of G_{up} . The figure a below shows the relation of figure 4a in the manuscript with data for which it was not possible to measure the tether diameter (blue open circles).

Additionally, there were frustrated uptake experiments, which it was not possible to complete the uptake process due to high membrane tensions (see figure c below).

a Uptake energy G_{up} as a function of Λ , including the uptake experiments for which were not possible to measure the formed tether after successful uptake. **b** Tether beyond optical resolution formed in tense vesicles. **c** Frustrated uptake experiment in tense vesicles. Narrower tether of the order of 100nm would be expected to be formed from **b** and **c**.

Also, membrane theory simply predicts that the tether has a radius determined by an interplay between bending rigidity and tension as: $r = \sqrt{\kappa / (2\sigma)}$. The data presented suggests that this widely tested relation is wrong. Is there any explanation for this discrepancy (beside the problem of resolution?).

Reply: Thanks to your careful reading, we realized that the tether diameter was used instead of the radius to calculate Λ . Hence, the Λ values in the abscissa of Figure 4 were too large by a factor of 2. Figure 4 is correct now. As described in the manuscript, we measured the tether force F_t by the optical tweezers and measured the tether radius R_t from the fluorescence intensity profile. Then we used the widely tested relations to obtain the membrane rigidity $K = \frac{1}{2\pi} F_t \cdot R_t$ and the tension $\sigma = \frac{1}{4\pi} F_t / R_t$ and of course $\Lambda = \sqrt{K/\sigma} = \sqrt{2}R_t$ as used for figure 4.

The use of this relation may not be fully adequate to get the mechanical properties of flaccid vesicles, since their tether tube is not as stable as the one formed in a tense vesicle. However, the goal here was to have a sorting tool for the data, and less to provide accurate values of mechanical properties of the vesicles.

Vesicle	Tether force F_t (pN)	Tether Radius R_t (μm)	Tension σ ($kT/\mu\text{m}^2$)	Bending rigidity K (kT)	Lambda Λ (μm)
Tense	2.1	0.14	300	12	0.2
Flaccid	0.17	0.23	15	1.6	0.33

13. Line 350: it is written that “thermal equilibrium was established”. Can the authors demonstrate this, e.g. by calculating tau = gamma / kappa and comparing that to the waiting time at each position?

Reply: Sure. The first sentence reads: “At each position and measurement (0.1 s, 100.000 data points), thermal equilibrium was established”. The waiting time is 0.1s, which is the time to reach equilibrium. By looking at figure 3d, one can see that the longest relaxation times are 0.2 ms for the tense GUV and 0.75 ms for the flaccid GUV. Hence, we give the system long enough time to equilibrate. We added the term dwell time to the value 0.1 s to make things clearer.

14. Line 437 / Fig 4B: this is a speculative statement, based on (as far as I can see) a single data point in Fig 4B with large error bars.

Reply: We assume that you are referring to figure 4a showing the decay of uptake energies with increasing flaccidness of vesicles (increasing Λ). There is lots of single data (shown by asterisk markers) and there is pooled data (shown by solid circles), i.e. not just a single point. Figure 4a shows by four solid circles that the uptake energy decays continuously from about $8 \cdot 1000 k_B T$, to about $4 \cdot 1000 k_B T$, to $1 \cdot 1000 k_B T$ and to $0.8 k_B T$ (for $\Lambda < 0.5$). We estimated the decay to be about eight-fold on average. In our opinion the statement is not speculative and as described above, there is lots of data indicating the decay in G_{up} .

RESPONSE TO REVIEWER 2:

Reviewer #2 (Remarks to the Author):

The manuscript of Ayala et al. comprehensively investigates the fluctuations of GUV membranes with different physical conditions and how those fluctuations affect the uptake of a fluctuating microscale particle. The authors make use of experimental techniques in conjunction with analytical theories that help explain their findings. The uptake of particles by membranes itself is an important research area, and it is exciting to see how optical tweezers are applied to visualize the process. Generally speaking, the overall significance of this work is high and the topic attracts a broad spectrum of researchers. However, to further improve the quality of the work and make a more substantial impact on the field, I would suggest that the authors carefully address the following issues.

Abstract:

1. In describing thermal fluctuations, the authors should be more specific. For example, in the sentence "... the uptake energy into a GUV becomes predictable since it increases with smaller and shorter thermal fluctuations", are these fluctuations smaller in amplitude and shorter in wavelength?

Reply: You are right, we can be more specific. Small fluctuations means small in amplitude, short fluctuations are meant to be short in lifetime, i.e. quickly decaying. We corrected and slightly rephrased the abstract to make the important points clearer.

2. The sentence "the reduced particle uptake energy using protein-ligand interactions LecA-Gb3 or Biotin-Streptavidin result also from stronger membrane fluctuations" does not seem to be correct, as I thought Biotin-Streptavidin interactions result in reduced fluctuations in the Discussion section.

Reply: Apparently, this misunderstanding results from the same ambiguous phrasings. As can be seen in figure 5, for LecA-Gb3 or Biotin-Streptavidin the uptake energy is reduced. This goes along with a reduced stiffness (more pronounced fluctuations, i.e. stronger in amplitude) and with a reduced friction (shorter relaxation times). We rephrased the abstract and the corresponding part in the discussion section:

"As a system interesting to infection biology, we investigated the interaction between DOPC lipid membrane enriched with the glycosphingolipid Gb3 (as receptor) and the bacterial lectin LecA, which was coated on the beads. In this case not only the uptake energies were lowered by an order of magnitude (blue curves), also the friction factors and stiffnesses were significantly reduced, which was also displayed in Fig. 5d. This means that the membrane fluctuations were larger in amplitude (lower stiffness) with shorter relaxation-times (lower friction) during the whole particle indentation and uptake."

Result:

3. The equations are not correctly numbered. All equations are mislabeled as equation (0).

Reply: This must have happened during the PDF conversion process on the Nature Communications website. We are totally sorry for this, since this must have turned the reading of the manuscript and the understanding of the theory extremely difficult.

4. The authors assume a linear superposition for both the stiffness and friction coefficient to add up the effects of bulk fluid and membrane in the presence of an optical trap. While the elastic term (stiffness) is additive, the viscous time (friction coefficient) typically is not. In fact, I would argue that the drag of the bead would depend on its distance from the membrane. Similarly, once the bead directly interacts (in contact or wrapped) with the membrane, the relaxation of the intervening fluid would make the particle-membrane response highly coupled, and the friction contributions non-additive. Can the authors verify this assumption?

Reply: You are mentioning two situations. We agree, that the drag of the bead decreases with the distance from the membrane, which we investigated in detail in (Andreas Meinel, 2014) and (Jünger, et al., 2015). In the second situation, when the bead is already in contact with the membrane, the fluid relaxation definitely makes the particle membrane response coupled, as you said. It is clear that the bulk friction γ_{bd} is increased by a perturbation term γ_{mem} , which changes for each distance d . On the other hand a decoupling between elasticity and friction is possible on very high frequencies ($\tau < 30 \mu s$, i.e. approximately $\omega \rightarrow \infty$).

I have rewritten and shortened the theory part, which should be clearer now. I see no obvious reason why the friction contributions should not add up as described by the eq. (5):

$$AC(\tau, d) \approx \frac{k_B T}{\kappa_{mem}(d) + \kappa_{opt}} - \frac{k_B T}{\gamma_{mem}(d) + \gamma_{bd}} \cdot \tau.$$

Nevertheless, we rephrased eq. (1) and provided some an additional explanation for our approximations.

5. How is q_{max} defined in this work compared with the intrinsic smallest fluctuation length scale of a free membrane? Can the authors justify their choice of the smallest length scale of the particle-membrane system being R_b ? Perhaps the physical conditions of GUVs should be taken into account in making the decision? For a thermally equilibrated membrane, the fluctuation-dissipation relation should be held for all the relevant undulation modes defined based on the smallest membrane length scale. I wondered whether this choice of length scale precludes other fluctuation modes that could impact the theoretical interpretation of the correlation time in this work.

Reply: In our case, the largest wave number $q_{max} = \pi/R_b$, depends on the radius R_b of the optically trapped bead, hence $q_{max} = 2\pi/\mu m$. In our theoretical/simulation results of figure 2, we displayed the maximal wave number $q_{10} = 10 \cdot q_0 = 10 \cdot \pi/R_g = 10 \cdot \pi/10\mu m = \pi/\mu m < q_{max} = 2\pi/\mu m$ for a good visibility. However, as already visible in figure 2 a,b,c, adding

modes $q > q_{10}$, does not change the friction or stiffness significantly, or in Figure 2c unexpectedly. The expected number of fluctuation modes is given by R_g / R_b , which is $N_{mx} = 15$ and $N_{mx} = 26$, respectively. The full range of modes is displayed in the new supplementary figure S4. You're right that the membrane parameters K and σ define the number N of modes which are necessary to be considered, i.e. set an upper limit $q_{max} = q_N$.

The largest wave number for free membranes is about the thickness of the membrane, which for our case is beyond experimental resolution.

6. Related to my previous point, an overview of the involved characteristic time and length scales (definitions and values) should be provided. As the particle-membrane problem is essentially a fluid dynamics problem, a discussion about how the authors distinguish between “short” and “long” time scales, “high” and “low” fluctuations (in terms of frequency?), as well as “long” and “short” wavelengths would be necessary. How are these scales or modes related to the correlation time? The authors mention that 30 μ s is short, but a comparison between this number and other relevant time scales (such as the fluid viscous dissipation time and the particle diffusion time) would make a strong point for fluid mechanists? I noticed that these terms are interchangeably used in the manuscript, and it would be good to have a more unified description for these terms to help the readers relate the observations presented in different figures.

Reply: We have inserted a table at the beginning of the results section summarizing the most important times and length scales, including the definitions for letters describing parameters. The discussion of the different temporal scales autocorrelation times has been extended, misleading or ambiguous phrasings have been corrected or improved. As explained in the text, the dwell time of the optical trap is $\Delta t = 100$ ms, such that 100,000 points can be recorded at a sampling rate of 1 MHz. Δt is long enough such that most membrane modes without bead can reach equilibrium. Single mode relaxation in contact with a 1 μ m bead is typically below 10 ms, with trap autocorrelation times of $\tau_c = \gamma_{bd}/\kappa_{opt} \approx 0.1$ ms ...2 ms. The 30 μ s analysis time to recover friction coefficients from free diffusion of a trapped bead with or without membrane contact are significantly lower than the autocorrelation times $\tau_{tot}(d) = 0.1...0.8$ ms, allowing reliable measurements of the bead's total viscous drag. We assume that the relaxation time of single membrane modes corresponds to the time the fluid needs to be displaced within the membrane deformations, which are the narrower the shorter the wavelength and the higher the mode. This explains the fact that the viscous drag increases with the mode number.

We finally went through all the figures to check whether terms have been used interchangeably, which should not be the case. Everything seems to be fine now.

7. The overall presentation of the notations can be improved. The authors should try to simplify the notations and make the necessary number of variables minimal. For example, y and h seem to be representing the membrane shape (deformation), but throughout the main text, h is probably used only once.

Reply: The only way to simplify the notation was to replace y by h , although h represents the membrane shape deformation, and y the indentation caused by the optical trap. We changed the notation in the main manuscript and supplementary material.

8. In line 192, should it be Figure 1A instead of Figure 1B?

Reply: The description in line 192 (previous version of the manuscript) refers also to Figure 1a. We corrected this.

9. The line colors in Figure 2A-2C should be consistent. In 2A the membrane-only and bead-only curves are in red and black, while in 2B the membrane-only curve is also in black. Perhaps the authors can change the colors for the bead-membrane curves instead of using blue and red.

Reply: We changed the line colors in Fig. 2a-c as you proposed.

10. It is not clear to me what the “Excluding, adding modes” blocks mean in Figure 2C. The green dashed block does not seem to correspond to the top green axis.

Reply: Yes, the axes were not well labeled. We improved figure 2c and fixed a typo in the programming code, which changed the slope of the curves slightly, but not significantly.

Now, the bottom axis describes how an increasing sum of modes ($1 \dots N_{mx}$) reduces the relaxation time, but also how a decreasing sum of modes ($n_0 \dots N_{mx}$) increases the relaxation time. E.g. for $n_0 = 7$, only the modes 8, 9, and 10 are added up, since the lower modes (1...6) are suppressed.

c Decay time of modes

Excluding and adding modes is further described by the formulas shown as insets. We adapted the colors from the previous figures for the smaller GUV with bead (light blue) and larger GUV with beads (blue).

In addition, a new supplementary figure S4 describes the continuous suppression of fluctuation modes for various membrane and trap parameters.

11. Can the authors more specifically define “ y ” in the section “Membrane deformation by a particle” for the adhesion energy G_{ad} ? I would have thought that the wrapped surface area of the particle would be approximately the total surface area ($4\pi R_b^2$) minus the circular area of the tube ($2\pi R_t$), instead of $2\pi R_b y$. This would

affect the determination of the uptake distance d_{up} and hence would deserve more explanation.

Reply: In principle your estimation for the contact area is useful and close to reality.

The adhesion area is assumed to be proportional to the contact area between the bead and the membrane. The contact area is defined as the surface area of a spherical cap wrapping the bead $A_{cap} = 2\pi R_b h_{loc}$ (see left side of figure S10). As the bead deforms the membrane, the contact area increases as a function of h_{loc} (see figure S11) until an equilibrium of local and global forces is reached. This is illustrated and expressed in figure S15 and in the following sentence in the Supplementary note 5 of the supplementary material:

“The indentation function $d(h_{lo})=h_{lo}+h_{gl}=h_{lo}+z(h_{lo})h_{lo}$ can be found by the zeros $z(h_{lo})$ of the corresponding force difference $F^{loc}(h_{lo})-F^{glo}=0$, such that $F^{loc}(h_{lo})=F^{glo}(z(h_{lo}))$.

Although the deformation forces can be derived analytically, the minimum $z(h_{lo})$ of the force difference is found numerically (root command). $z(h_{lo})$ provides the ratio of the global indentation h_{gl} relative to the local indentation h_{lo} .

It can be seen at the bottom right of Figure S 15 that the total indentation $d(h_{lo})$ varies around $d(h_{lo}) \approx 2h_{lo}$ for a variety of membrane parameters.”

12. In line 360, should it be “horizontal dotted lines” in Figure 3B?

Reply: In this case the phrase “horizontal dotted lines” refers to the distance $d = d_{up}$ not to the maximal force. There are only two vertical dotted lines in each graph. One refers to the distance $d = 0$ at which contact between the membrane and the particle occurs, the other refers to the distance at which maximal force is reached named as $d = d_{up}$.

In order to avoid confusion, we wrote in the second paragraph of subsection “Measured profiles for force and energy during particle uptake”:

“When a maximum area of the particle is wrapped by the membrane, the force becomes maximal at the uptake distance $d = d_{up}$ (second vertical dotted line in Fig. 3b).”

13. The authors write, “...to achieve reasonable values, not all modes up to N_{mx} over the distance d_{up} were suppressed, but over the distance $2d_{up}$...” What does this mean? Is there a systematic or rational way of determining the decreasing rate of the number of fluctuation modes?

Reply: You have read the text very carefully, I'm impressed. And no, I'm not aware of a systematic way of determining this rate. We have improved the text as follows and added another sentence pointing out the necessity to develop a more advanced theory in the future, see the second paragraph in subsection “Measured profiles for fluctuations parameters during particle uptake”:

“However, to achieve friction values comparable to the experiments, not all modes up to N_{mx} over the distance d_{up} were suppressed, but over the distance $2 \cdot d_{up}$ in the denominator in eq.(7). The mode suppression rate with d needs to be investigated in more detail in the future.”

14. How are the three calculated curves determined in Figure 4? Can the authors specify the equation numbers or remind the readers of expressions for those theoretical curves?

Reply: Okay. Since the mechanical properties of the GUV membranes can change with all combinations of bending rigidity K and surface tension σ , we used K as parameter (three curves) and varied the surface tension from $\sigma_{max} = 400 \text{ k}_B\text{T}/\mu\text{m}^2$ and $\sigma_{min} = 15 \text{ k}_B\text{T}/\mu\text{m}^2$. The following text was added in the legend of figure 4b:

“Theory curves $G_{up}(\sigma)$ and $F_{up}(\sigma)$ are plotted against $\Lambda(\sigma)$ and are shown for bending rigidities $K=2.4, 7.2$ and $18 \text{ k}_B\text{T}$ in black, red and orange.”

In Figure 4, the Λ values in the abscissa of the graphs were corrected by a factor of $\frac{1}{2}$, which does not change our results.

15. What are the membrane surface densities and bead coating densities of those functional groups in the section of adhesion-mediated particle uptake? A correlation between those grafting densities and G_{up} , γ_{up} , and κ_{up} should be given. This information would be important for the theoreticians to compare the results in Figure 5.

Reply: The required information was added to Material and Methods section in the part referring to “LecA coating”. It now reads:

“The linker densities for Gb3-DOPC/LecA and biotinylated - DOPC/Streptavidin experiments showed in Figure 5 are approximately $607 \mu\text{m}^{-2}$ and $450 \mu\text{m}^{-2}$, respectively. They were obtained from the detachment energy measured from pulling experiments in our previous work (Ramin Omidvar, 2021).”

However, in the present work we did not vary the concentrations of the linkers (LecA-Gb3, Biotin-Streptavidin) to see how the linker densities influence the uptake energy (G_{up}), and the maximal value of viscous drag (γ_{up}) and elasticity κ_{up} . A further study would be needed to establish a correlation between the parameters presented in Figure 5 and different linker densities.

Discussion:

16. In Figure 2C, it is shown that τ_c decreases if more higher modes (N) are included. However, in Figure 2D, the autocorrelation function with more modes included exhibits a slower decay, suggesting the opposite trend. Can the authors clarify this?

Reply: Thank you for this hint, the colors orange and green were exchanged in the legend of figure 2d. Now it should be fine.

17. Can the authors also explain how an increase in relaxation time corresponds to an increase in friction in Figure 2C (statement made in line 591)? This would seem contradictory to the argument made in Figure 5 that “the membrane fluctuations are large in amplitude (low stiffness) with long relaxation times (low friction)”.

Reply: I think everything around figure 2c is correct. The friction and autocorrelation time decrease when higher modes are added (increasing N, dashed lines), but friction and AC time increase over a considerable range when lower modes are suppressed (increasing n0, solid lines). Both are shown in figure 2c and in a new supplementary figure S4.

We improved the sentence at the end of the third paragraph of “Can higher mode fluctuations make a membrane stiffer and more viscous?” in Discussion section as follows:

“This corresponds to the effect that low n modes, i.e. long wavelength modes are subsequently suppressed. Exactly this suppression effect leads now to an increase in relaxation time or friction as shown by the solid lines in Fig. 2c. “

There was a typo error in the discussion of Fig. 5d in the second paragraph of subsection “Strong adhesion energies suppress membrane fluctuations, while LecA-Gb3 interactions increase them”, where “long” should be replaced by “short”. This error was also detected by reviewer 1. Now the sentence reads:

“...also the friction factors and stiffnesses were significantly reduced, which was also displayed in Fig. 5d. This means that the membrane fluctuations were larger in amplitude (lower stiffness) with shorter relaxation-times (lower friction) during the whole particle indentation and uptake.”

Supplementary:

18. The notations in supplementary text 4 (in terms of h) are inconsistent with the main text (in terms of y), which is confusing. Meanwhile, I find that a lot of the sentences here (page 4 of supplementary) are redundant, as they have been repeated in the main text.

Reply: If you mean the expressions for $G_{mem}(h_{lo}+h_{gl})$ and $G_{mem}(y_{loc}+y_{glo})$, I would think there is no inconsistency, but the variables have different names. I agree that this is not helpful to have. We moved the descriptions of G_{mem} to the supplementary material and now there should be no y dependency of quantities in the main text.

19. I find the presentations of Figures 3-4, 6-8, 10-17 difficult to digest. They seem to

include the source code of a specific software called Mathcad. It would be helpful if the authors can provide paragraphs explaining more details of the figures.

Reply: We replaced the MathCad screenshot formulas by MathType formulas as they are in the text, so now the explanations given in the text are readily identified in the supplementary figures. Also, we wrote a little bit more of text explaining the supplementary figures.

References

- Andreas Meinel, B. T. (2014). Induced phagocytic particle uptake into a giant unilamellar vesicle. *Soft Matter*, 10, 3667-3678.
- Bolognesi, G. F.-R. (2018). Sculpting and fusing biomimetic vesicle networks using optical tweezers. *Nat Commun*, 9, 1882.
- Català, F. M.-U. (2017). Influence of experimental parameters on the laser heating of an optical trap. *Sci Rep*, 7, 16052.
- Debjit Roy, J. S. (2020). Mechanical Tension of Biomembranes Can Be Measured by Super Resolution (STED) Microscopy of Force-Induced Nanotubes. *Nano Lett.*, 20, 5, 3185-3191.
- Dimova, R. D. (2014). Inward and outward membrane tubes pulled from giant vesicles. *Journal of Physics D: Applied Physics*, 47, 28.
- Friddin, M. B.-R. (2019). Direct manipulation of liquid ordered lipid membrane domains using optical traps. *Commun Chem*, 2, 6.
- Jünger, F., Kohler, F., Meinel, A., Meyer, T., Nitschke, R., Erhard, B., et al. (2015). Measuring Local Viscosities near Plasma Membranes of Living Cells with Photonic Force Microscopy. *Biophysical Journal*, 109, 5.
- Ramin Omidvar, Y. A. (2021). Quantification of nanoscale forces in lectin-mediated bacterial attachment and uptake into giant liposomes. *Nanoscale*, 13, 4016.
- Renner M, D. Y. (2011). Lateral Diffusion on Tubular Membranes: Quantification of Measurements Bias. *PLoS ONE*, 6 (9).
- Z. V. Leonenko, E. F. (2004). Investigation of Temperature-Induced Phase Transitions in DOPC and. *Biophysical Journal*, 86.

Reviewers' Comments:

Reviewer #1:

Remarks to the Author:

I would like to thank the authors for the detailed answers and the changes made to the manuscript. I am justified with the answers and the revised version. Personally, I would have expressed some things in a different way, but it is absolutely in the hands of the authors to decide on notation and the level of theory they deem necessary.

Therefore, I fully support publication and look forward seeing this beautiful work published in Nature Communications.

Reviewer #2:

Remarks to the Author:

I thank the authors for taking the time to answer my questions and revise the manuscript. My questions have been addressed carefully, and I only have one minor suggestion. In Table 1, if the ellipsis between numbers (e.g., 0...8) represents the range of numbers, then it should be replaced by the en dash.